# Graph Neural Networks for Link Prediction with Subgraph Sketching

**Benjamin P. Chamberlain** [*†‡]
Charm Therapeutics

**Sergey Shirobokov** [*†]
ShareChat AI

**Emanuele Rossi** [†]
Imperial College London

**Fabrizio Frasca** [†]
Imperial College London

**Thomas Markovich**
Twitter Inc.

**Nils Hammerla**
Twitter Inc.

**Michael M. Bronstein** [†]
University of Oxford

**Max Hansmire**
Twitter Inc.

## Abstract

Many Graph Neural Networks (GNNs) perform poorly compared to simple heuristics on Link Prediction (LP) tasks. This is due to limitations in expressive power such as the inability to count triangles (the backbone of most LP heuristics) and because they can not distinguish automorphic nodes (those having identical structural roles). Both expressiveness issues can be alleviated by learning link (rather than node) representations and incorporating structural features such as triangle counts. Since explicit link representations are often prohibitively expensive, recent works resorted to subgraph-based methods, which have achieved state-of-the-art performance for LP, but suffer from poor efficiency due to high levels of redundancy between subgraphs. We analyze the components of subgraph GNN (SGNN) methods for link prediction. Based on our analysis, we propose a novel full-graph GNN called ELPH (Efficient Link Prediction with Hashing) that passes subgraph sketches as messages to approximate the key components of SGNNs without explicit subgraph construction. ELPH is provably more expressive than Message Passing GNNs (MPNNs). It outperforms existing SGNN models on many standard LP benchmarks while being orders of magnitude faster. However, it shares the common GNN limitation that it is only efficient when the dataset fits in GPU memory. Accordingly, we develop a highly scalable model, called BUDDY, which uses feature precomputation to circumvent this limitation without sacrificing predictive performance. Our experiments show that BUDDY also outperforms SGNNs on standard LP benchmarks while being highly scalable and faster than ELPH.

## 1 Introduction

Link Prediction (LP) is an important problem in graph ML with many industrial applications. For example, recommender systems can be formulated as LP; link prediction is also a key process in drug discovery and knowledge graph construction. There are three main classes of LP methods: (i) **heuristics** (See Appendix C.1) that estimate the distance between two nodes (e.g. personalized page rank (PPR) (Page et al., 1999) or graph distance (Zhou et al., 2009)) or the similarity of their neighborhoods (e.g Common Neighbors (CN), Adamic-Adar (AA) (Adamic & Adar, 2003), or Resource Allocation (RA) (Zhou et al., 2009)); (ii) **unsupervised node embeddings** or **factorization** methods, which encompass the majority of production recommendation systems (Koren et al., 2009; Chamberlain et al., 2020); and, recently, (iii) **Graph Neural Networks**, in particular of the Message-Passing type (MPNNs) (Gilmer et al., 2017; Kipf & Welling, 2017; Hamilton et al., 2017).[1] GNNs excel in graph- and node-level tasks, but often fail to outperform node embeddings or heuristics on common LP benchmarks such as the Open Graph Benchmark (OGB) (Hu et al., 2020).

There are two related reasons why MPNNs tend to be poor link predictors. Firstly, due to the equivalence of message passing to the Weisfeiler-Leman (WL) graph isomorphism test (Xu et al., 2019; Morris et al., 2019), standard MPNNs are provably incapable of counting trian-

---

[*]Equal contribution. [†]Work done while at Twitter Inc. [‡]benjamin.chamberlain@gmail.com.

[1]GNNs are a broader category than MPNNs. Since the majority of GNNs used in practice are of the message passing type, we will use the terms synonymously.

gles (Chen et al., 2020) and consequently of counting Common Neighbors or computing one-hop or two-hop LP heuristics such as AA or RA. Secondly, GNN-based LP approaches combine permutation-equivariant structural node representations (obtained by message passing on the graph) and a readout function that maps from two node representations to a link probability. However, generating link representations as a function of equivariant node representations encounters the problem that all nodes $u$ in the same orbit induced by the graph automorphism group have equal representations. Therefore, the link probability $p(u, v)$ is the same for all $u$ in the orbit independent of e.g. the graph distance $d(u, v)$ (Figure 1). (Srinivasan & Ribeiro, 2019).

This is a result of GNN's built-in permutation equivariance, which produces equal representations for any nodes whose enclosing subgraphs (corresponding to the receptive field of the GNN) are isomorphic.[2] We refer to this phenomenon as the *automorphic node problem* and define *automorphic nodes* (denoted $u \cong v$) to be those nodes that are indistinguishable by means of a given $k$-layer GNN. On the other hand, transductive node embedding methods such as TransE (Bordes et al., 2013) and DeepWalk (Perozzi et al., 2014), or matrix factorization (Koren et al., 2009) do not suffer from this problem as the embeddings are not permutation equivariant.

Figure 1: Nodes 2 and 4 are in the same orbit induced by the graph's automorphism group. As a result, a conventional GNN will assign the same probability to links (1,2) and (1,4).

Several methods have been proposed to improve GNN expressivity for LP. Most simply, adding unique node IDs immediately makes all structural node representations distinguishable, but at the expense of generalization (Abboud et al., 2021) and training convergence (Sato et al., 2021). Substructure counts may act as permutation-equivariant approximately unique identifiers (Bouritsas et al., 2022), but they require a precomputation step which may be computationally intractable in the worst case. More successfully, a family of structural features, sometimes referred to as *labeling tricks*, have recently been proposed that solve the automorphic node problem while still being equivariant and having good generalization (Li et al., 2020; Zhang et al., 2021; You et al., 2021). However, adding structural features amounts to computing structural node representations that are conditioned on an edge and so can no longer be efficiently computed in parallel. For the purpose of tractability, state-of-the-art methods for LP restrict computation to subgraphs enclosing a link, transforming link prediction into *binary subgraph classification* (Zhang et al., 2021; Zhang & Chen, 2018; Yin et al., 2022). Subgraph GNNs (SGNN) are inspired by the strong performance of LP heuristics compared to more sophisticated techniques and are motivated as an attempt to learn data-driven LP heuristics.

Despite impressive performance on benchmark datasets, SGNNs suffer from some serious limitations: (i) Constructing the subgraphs is expensive; (ii) Subgraphs are irregular and so batching them is inefficient on GPUs (iii); Each step of inference is almost as expensive as each training step because subgraphs must be constructed for every test link. These drawbacks preclude many applications, where scalability or efficient inference are required.

**Main contributions.** (i) We analyze the relative contributions of SGNN components and reveal which properties of the subgraphs are salient to the LP problem. (ii) Based on our analysis, we develop an MPNN (ELPH) that passes subgraph sketches as messages. The sketches allow the most important qualities of the subgraphs to be summarized in the nodes. The resulting model removes the need for explicit subgraph construction and is a full-graph MPNN with the similar complexity to GCN. (iii) We prove that ELPH is strictly more expressive than MPNNs for LP and that it solves the automorphic node problem. (iv) As full-graph GNNs suffer from scalability issues when the data exceeds GPU memory, we develop BUDDY, a highly scalable model that precomputes sketches and node features. (v) We provide an open source Pytorch library for (sub)graph sketching that generates data sketches via message passing on the GPU. Experimental evaluation shows that our methods compares favorably to state-of-the-art both in terms of accuracy and speed.

## 2 PRELIMINARIES

**Notation.** Let $G = (\mathcal{V}, \mathcal{E})$ be an undirected graph comprising the set of $n$ nodes (vertices) $\mathcal{V}$ and $e$ links (edges) $\mathcal{E}$. We denote by $d(u, v)$ the *geodesic distance* (shortest walk length) between nodes

---

[2]More precisely, WL-equivalent, which is a necessary but insufficient condition for isomorphism.

$u$ and $v$. Let $S = (\mathcal{V}_S \subseteq \mathcal{V}, \mathcal{E}_S \subseteq \mathcal{E})$ be a node-induced subgraph of $G$ satisfying $(u, v) \in \mathcal{E}_S$ iff $(u, v) \in \mathcal{E}$ for any $u, v \in \mathcal{V}_S$. We denote by $S_{uv}^k = (\mathcal{V}_{uv}, \mathcal{E}_{uv})$ a $k$-hop subgraph enclosing the link $(u, v)$, where $\mathcal{V}_{uv}$ is the union of the $k$-hop neighbors of $u$ and $v$ and $\mathcal{E}_{uv}$ is the union of the links that can be reached by a $k$-hop walk originating at $u$ and $v$ (for simplicity, where possible, we omit $k$). Similarly, $S_u^k$ is the $k$-hop subgraph enclosing node $u$. The given features of nodes $\mathcal{V}_{uv}$ are denoted by $\mathbf{X}_{uv}$ and the derived structure features by $\mathbf{Z}_{uv}$. The probability of a link $(u, v)$ is denoted by $p(u, v)$. When nodes $u$ and $v$ have isomorphic enclosing subgraphs (i.e., $S_u \cong S_v$), we write $u \cong v$.

**Sketches for Intersection Estimation.** We use two sketching techniques, *HyperLogLog* (Flajolet et al., 2007; Heule et al., 2013) and *MinHashing* (Broder, 1997). Given sets $\mathcal{A}$ and $\mathcal{B}$, *HyperLogLog* efficiently estimates the cardinality of the union $|\mathcal{A} \cup \mathcal{B}|$ and *MinHashing* estimates the Jaccard index $J(\mathcal{A}, \mathcal{B}) = |\mathcal{A} \cap \mathcal{B}| / |\mathcal{A} \cup \mathcal{B}|$. We combine these approaches to estimate the intersection of node sets produced by graph traversals (Pascoe, 2013). These techniques represent sets as sketches, where the sketches are much smaller than the sets they represent. Each technique has a parameter $p$ controlling the trade-off between the accuracy and computational cost. Running times for merging two sets, adding an element to a set, and extracting estimates only depend on $p$, but they are constant with respect to the size of the set. Importantly, the sketches of the union of sets are given by permutation-invariant operations (element-wise $\min$ for minhash and element-wise $\max$ for hyperloglog). More details are provided in Appendix C.3.

**Graph Neural Networks for Link Prediction.** Message-passing GNNs (MPNNs) are parametric functions of the form $\mathbf{Y} = \mathrm{GNN}(\mathbf{X})$, where $\mathbf{X}$ and $\mathbf{Y}$ are matrix representations (of size $n \times d$ and $n \times d'$, where $n$ is the number of nodes and $d, d'$ are the input and output dimensions, respectively) of input and output node features. *Permutation equivariance* implies that $\mathbf{\Pi}\mathrm{GNN}(\mathbf{X}) = \mathrm{GNN}(\mathbf{\Pi}\mathbf{X})$ for any $n \times n$ node permutation matrix $\mathbf{\Pi}$. This is achieved in GNNs by applying a local permutation-invariant aggregation function $\square$ (typically sum, mean, or max) to the neighbor features of every node ('message passing'), resulting in a node-wise update of the form

$$\mathbf{y}_u = \gamma \left( \mathbf{x}_u, \square_{v \in \mathcal{N}(u)} \phi \left( \mathbf{x}_u, \mathbf{x}_v \right) \right), \tag{1}$$

where $\phi, \gamma$ are learnable functions. MPNNs are upperbounded in their discriminative power by the Weisfeiler-Leman isomorphism test (WL) (Weisfeiler & Leman, 1968), a procedure iteratively refining the representation of a node by hashing its star-shaped neighborhood. As a consequence, since WL always identically represents automorphic nodes ($u \cong v$), any MPNN would do the same: $\mathbf{y}_u = \mathbf{y}_v$. Given the node representations $\mathbf{Y}$ computed by a GNN, link probabilities can then be computed as $p(u, v) = R(\mathbf{y}_u, \mathbf{y}_v)$, where $R$ is a learnable readout function with the property that $R(\mathbf{y}_u, \mathbf{y}_v) = R(\mathbf{y}_u, \mathbf{y}_w)$ for any $v \cong w$. This node automorphism problem is detrimental for LP as $p(u, v) = p(u, w)$ if $v \cong w$ even when $d(u, v) \gg d(u, w)$. As an example, in Figure 1 $2 \cong 4$, therefore $p(1, 2) = p(1, 4)$ while $d(1, 2) = 2 < d(1, 4) = 3$. As a result, a GNN may suggest to link totally unrelated nodes ($v$ may even be in a separate connected component to $u$ and $w$, but still have equal probability of connecting to $u$ (Srinivasan & Ribeiro, 2019)).

**Subgraph GNNs (SGNN).** (Zhang et al., 2021; Zhang & Chen, 2018; Yin et al., 2022) convert LP into binary graph classification. For a pair of nodes $u, v$ and the enclosing subgraph $S_{uv}$, SGNNs produce node representations $\mathbf{Y}_{uv}$ and one desires $R(\mathbf{Y}_{uv}) = 1$ if $(u, v) \in \mathcal{E}$ and zero otherwise. In order to resolve the automorphic node problem, node features are augmented with structural features (Bouritsas et al., 2022) that improve the ability of networks to count substructures (Chen et al., 2020). SGNNs were originally motivated by the strong performance of heuristics on benchmark datasets and attempted to learn generalized heuristics. When the graph is large it is not tractable to learn heuristics over the full graph, but global heuristics can be well approximated from subgraphs that are augmented with structural features with an approximation error that decays exponentially with the number of hops taken to construct the subgraph (Zhang & Chen, 2018).

## 3 ANALYZING SUBGRAPH METHODS FOR LINK PREDICTION

SGNNs can be decomposed into the following steps: (i) subgraph extraction around every pair of nodes for which one desires to perform LP; (ii) augmentation of the subgraph nodes with structure features; (iii) feature propagation over the subgraphs using a GNN, and (iv) learning a graph-level readout function to predict the link. Steps (ii)–(iv) rely on the existence of a set of subgraphs (i),

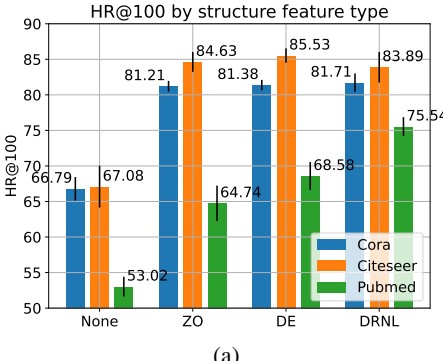 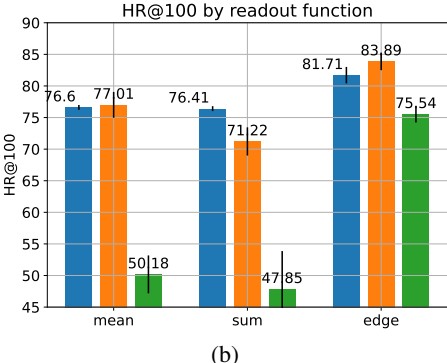

(a)                       (b)

Figure 2: (a) Structure features in SGNNs. (b) LP readout function over the output of all nodes in $S_{uv}$ (sum or mean) or just $u$ and $v$ (edge).

which is either constructed on the fly or as a preprocessing step. In the remainder of this section we discuss the inherent complexity of each of these steps and perform ablation studies with the goal of understanding the relative importance of each.

**Structure Features** Structure features address limitations in GNN expressivity stemming from the inherent inability of message passing to distinguish automorphic nodes. In SGNNs, permutation-equivariant distances $d(u, i)$ and $d(v, i) \forall i \in \mathcal{V}_{uv}$ are used. The three most well known are Zero-One (ZO) encoding (You et al., 2021), Double Radius Node Labeling (DRNL) (Zhang & Chen, 2018) and Distance Encoding (DE) (Li et al., 2020). To solve the automorphic node problem, all that is required is to distinguish $u$ and $v$ from $\mathcal{V}_{uv} \setminus \{u, v\}$, which ZO achieves with binary node labels. DRNL has $z_u = z_v = 1$ and $z_j = f(d(u, j), d(v, j)) > 1$, where $f : \mathbb{N}^2 \to \mathbb{N}$ is a bijective map. Distance Encoding (DE) generalizes DRNL; each node is encoded with a tuple $z_j = (d(u, j), d(v, j))$ (See Figure 5). Both of these

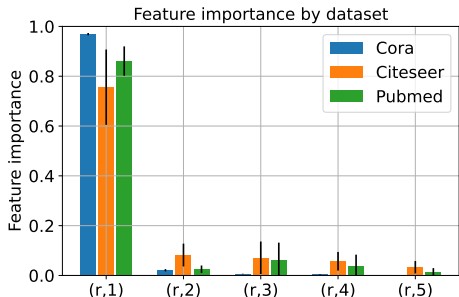

Figure 3: The importance of DRNL structure features. Importance is based on the weights in a logistic regression model using all DRNL features without node features.

schemes therefore include a unique label for triangles / common neighbors (See Appendix C.4 for more details on labeling schemes). The relative performance of ZO, DRNL, DE and no structure features is shown in Figure 2a. The use of structure features greatly improves performance and DRNL and DE slightly outperform ZO, with a more pronounced outperformance for the larger Pubmed dataset. It is important to note that structure features are conditioned on edges and so can not be easily paralellized in the same way that node features usually are in GNNs and must be calculated for each link at both training and inference time. In particular DRNL requires two full traversals of each subgraph, which for regular graphs is $\mathcal{O}(deg^k)$, but for complex networks becomes $O(|\mathcal{E}|)$.

In Figure 3 we investigate the relative importance of DRNL structure features. Feature importance is measured using the weights in a logistic regression link prediction model with only structure feature counts as inputs. We then sum up feature importances corresponding to different max distance $r$ and normalize the total sum to one. The Figure indicates that most of the predictive performance is concentrated in low distances.

**Propagation / GNN** In SGNNs, structure features are usually embedded into a continuous space, concatenated to any node features and propagated over subgraphs. While this procedure is necessary for ZO encodings, it is less clear why labels that precisely encode distances and are directly comparable as distances should be embedded in this way. Instead of passing embedded DE or DRNL structure features through a GNN, we fixed the number of DE features by setting the max distance to three and trained an MLP directly on the counts of these nine features (e.g. (1,1): 3, (1,2): 1, etc.). Figure 4a shows that doing so, while leaving ceteris paribus (node features still propagated over the subgraph) actually improves performance in two out of three datasets. We also investigated if any features require SGNN propagation by passing both raw node features and structure feature counts

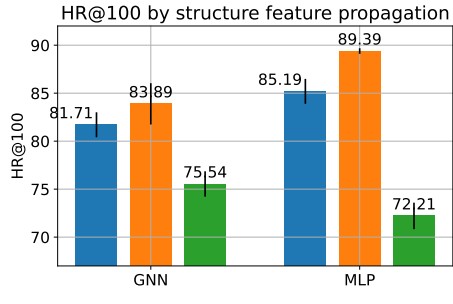
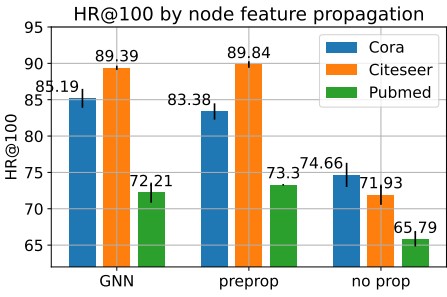

(a) Propagating structure features

(b) Propagating node features.

Figure 4: Examining the extent to which an SGNN is needed to propagate features. (a) An MLP on structure feature counts outperforms SGNN propagation in two of three datasets. (b) An MLP on structure feature counts and raw node features performs poorly (no prop), but performance can be recovered if node features are propagated over the graph in preprocessing (preprop).

through an MLP. The results in the right columns of Figure 4b indicate that this reduces performance severely, but by pre-propagating the features as $\mathbf{x}'_u = 1/|N(u)| \sum_{i \in N(u)} \mathbf{x}_i$ (middle columns) it is possible to almost recover the performance of propagation with the SGNN (left columns).

**Readout / Pooling Function**  Given SGNN node representations $\mathbf{Y}_{uv}$ on the subgraph, a readout function $R(S_{uv}, \mathbf{Y}_{uv})$ maps a representations to link probabilities. For graph classification problems, this is most commonly done by pooling node representations (graph pooling) typically with a mean or sum operation plus an MLP. For LP, an alternative is edge pooling with $R(\mathbf{y}_u, \mathbf{y}_v)$, usually with the Hadamard product. A major advantage of edge pooling is that it can be formulated subgraph free. Figure 2b indicates that edge pooling produces better predictive performance than either mean or sum pooling across all nodes in $\mathcal{V}_{uv}$.

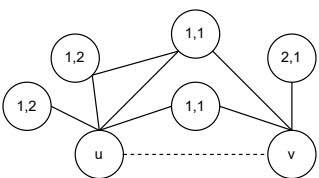

Figure 5: The DE node labeling scheme for link $(u, v)$

**Analysis Summary**  The main results of Section 3 are that (i) The inclusion of structure features leads to very large improvements across all datasets (Figure 2a); (ii) The processing of these features, by embedding them and propagating them with an SGNN is sub-optimal both in terms of efficiency and performance (Figure 4a); (iii) Most of the importance of the structure features is located in the lowest distances (Figure 3); and (iv) edge level readout functions greatly outperform mean or sum pooling over subgraphs (Figure 2b). If, on one hand, subgraphs are employed as a tractable alternative to the full graph for each training edge, on the other, generating them remains an expensive operation ($\mathcal{O}(deg^k)$ time complexity for regular graphs and $\mathcal{O}(|\mathcal{E}|)$ for complex networks with power law degree distributions [3], see Appendix C.2). Within this context, our analysis shows that if the information necessary to compute structure features for an edge can be encoded in the nodes, then it is possible to recover the predictive performance of SGNNs without the cost of generating a different subgraph for each edge. We build upon this observation to design an efficient yet expressive model in Section 4.

## 4  LINK PREDICTION WITH SUBGRAPH SKETCHING

We now develop a full-graph GNN model that uses node-wise subgraph sketches to approximate structure features such as the counts of DE and DRNL labels, which our analysis indicated are sufficient to encompass the salient patterns governing the existence of a link (Figure 4a).

### 4.1  STRUCTURE FEATURES COUNTS

Let $\mathcal{A}_{uv}[d_u, d_v]$ be the number of $(d_u, d_v)$ labels for the link $(u, v)$, which is equivalent to the number of nodes at distances exactly $d_u$ and $d_v$ from $u$ and $v$ respectively (See Figure 5). We compute

---

[3]Subgraphs can be pre-computed, but the subgraphs combined are much larger than the original dataset, exceeding available memory for even moderately-sized datasets.

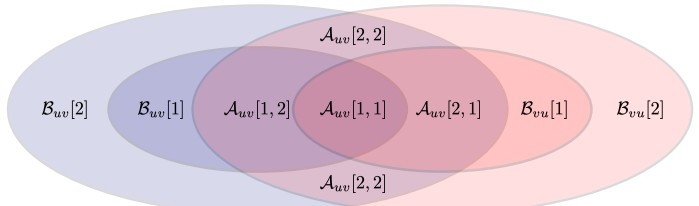

Figure 6: Blue and red concentric circles indicate 1 and 2-hop neighborhoods of $u$ and $v$ respectively. Structure features $\mathcal{A}$ and $\mathcal{B}$ measure the cardinalities of intersections of these neighborhoods.

$\mathcal{A}_{uv}[d_u, d_v]$ for all $d_u$, $d_v$ less than the receptive field $k$, which guarantees a number of counts that do not depend on the graph size and mitigates overfitting. To alleviate the loss of information coming from a fixed $k$, we also compute $\mathcal{B}_{uv}[d] = \sum_{d_v=k+1}^{\infty} \mathcal{A}_{uv}[d, d_v]$, counting the number of nodes at distance $d$ from $u$ and at distance $> k$ from $v$. We compute $\mathcal{B}_{uv}[d]$ for all $1 \leq d \leq k$. Figure 6 shows how $\mathcal{A}$ and $\mathcal{B}$ relate to the neighborhoods of the two nodes. Overall, this results in $k^2$ counts for $\mathcal{A}$ and $2k$ counts for $\mathcal{B}$ ($k$ for the source and $k$ for the destination node), for a total of $k(k+2)$ count features. These counts can be computed efficiently without constructing the whole subgraph for each edge. Defining $N_{d_u,d_v}(u,v) \triangleq N_{d_u}(u) \cap N_{d_v}(v)$, we have

$$\mathcal{A}_{uv}[d_u, d_v] = |N_{d_u,d_v}(u,v)| - \sum_{x \leq d_u, y \leq d_v, (x,y) \neq (d_u,d_v)} |N_{x,y}(u,v)| \tag{2}$$

$$\mathcal{B}_{uv}[d] = |N_d(u)| - \mathcal{B}_{uv}[d-1] - \sum_{i=1}^{d} \sum_{j=1}^{d} \mathcal{A}_{uv}[i,j] \tag{3}$$

where $N_d(u)$ are the $d$-hop neighbors of $u$ (i.e. nodes at distance $\leq d$ from $u$).

**Estimating Intersections and Cardinalities** $|N_{d_u,d_v}(u,v)|$ and $|N_d(u)|$ can be efficiently approximated with the sketching techniques introduced in section 2. Let $\mathbf{h}_u^{(d)}$ and $\mathbf{m}_u^{(d)}$ be the *HyperLogLog* and *MinHash* sketches of node $u$'s $d$-hop neighborhood, obtained recursively from the initial node sketches $\mathbf{h}_u^{(0)}$ and $\mathbf{m}_u^{(0)}$ using the relations $\mathbf{m}_u^{(d)} = \min_{v \in \mathcal{N}(u)} \mathbf{m}_v^{(d-1)}$ and $\mathbf{h}_u^{(d)} = \max_{v \in \mathcal{N}(u)} \mathbf{h}_v^{(d-1)}$, where $\min$ and $\max$ are elementwise. We can approximate the intersection of neighborhood sets as

$$|N_{d_u,d_v}(u,v)| \triangleq |N_{d_u}(u) \cap N_{d_v}(v)| = \tag{4}$$

$$= J(N_{d_u}(u), N_{d_v}(v)) \cdot |N_{d_u}(u) \cup N_{d_v}(v)| \tag{5}$$

$$\approx H\left(\mathbf{m}_u^{(d_u)}, \mathbf{m}_v^{(d_v)}\right) \cdot card\left(\max(\mathbf{h}_u^{(d_u)}, \mathbf{h}_v^{(d_v)})\right), \tag{6}$$

where $H(x,y) = 1/n \sum_i^n \delta_{x_i,y_i}$ is the Hamming similarity, *card* is the *HyperLogLog* cardinality estimation function (see Appendix C.3.1) and $\max$ is taken elementwise. *MinHashing* approximates the Jaccard similarity ($J$) between two sets and *HyperLogLog* approximates set cardinality. $|N_d(u)|$ can be approximated as $|N_d(u)| \approx card(\mathbf{h}_u^{(d)})$. The cost of this operation only depends on additional parameters of *HyperLogLog* and *MinHashing* (outlined in more detail in Section 2) which allow for a tradeoff between speed and accuracy, but is nevertheless independent of the graph size. Using this approximations we obtain estimates $\hat{\mathcal{A}}$ and $\hat{\mathcal{B}}$ of the structure features counts.

### 4.2 Efficient Link Prediction with Hashes (ELPH)

We present ELPH, a novel MPNN for link prediction. In common with other full-graph GNNs, it employs a feature propagation component with a link-level readout function. However, by augmenting the messages with subgraph sketches, it achieves higher expressiveness for the same asymptotic complexity. ELPH's feature propagation fits the standard MPNN formalism (Gilmer et al., 2017):

$$\mathbf{m}_u^{(l)} = \min_{v \in \mathcal{N}(u)} \mathbf{m}_v^{(l-1)}, \qquad \mathbf{h}_u^{(l)} = \max_{v \in \mathcal{N}(u)} \mathbf{h}_v^{(l-1)} \tag{7}$$

$$\mathbf{e}_{u,v}^{(l)} = \{\hat{\mathcal{B}}_{uv}[l], \hat{\mathcal{A}}_{uv}[d_u, l], \hat{\mathcal{A}}_{uv}[l, d_v] : \forall d_u, d_v < l\} \tag{8}$$

$$\mathbf{x}_u^{(l)} = \gamma^{(l)}\left(\mathbf{x}_u^{(l-1)}, \square_{v \in \mathcal{N}(u)} \phi^{(l)}\left(\mathbf{x}_u^{(l-1)}, \mathbf{x}_v^{(l-1)}, \mathbf{e}_{u,v}^{(l)}\right)\right) \tag{9}$$

where $\phi, \gamma$ are learnable functions, $\square$ is a local permutation-invariant aggregation function (typically sum, mean, or max), $\mathbf{x}_u^{(0)}$ are the original node features, and $\mathbf{m}_u^{(0)}$ and $\mathbf{h}_u^{(0)}$ are the *MinHashing* and *HyperLogLog* single node sketches respectively. Minhash and hyperloglog sketches of $N_l(u)$ are computed by aggregating with $\min$ and $\max$ operators respectively the sketches of $N_{l-1}(u)$ (Eq. 7). These sketches can be used to compute the intersection estimations $\hat{\mathcal{B}}_{uv}$ and $\hat{\mathcal{A}}_{uv}$ up to the $l$-hop neighborhood, which are then used as edge features (Eq. 8). Intuitively, the role of the edge features is to modulate message transmission based on local graph structures, similarly to how attention is used to modulate message transmission based on feature couplings. A link predictor is then applied to the final node representations of the form

$$p(u, v) = \psi\left(\mathbf{x}_u^{(k)} \odot \mathbf{x}_v^{(k)}, \{\hat{\mathcal{B}}_{uv}[d], \hat{\mathcal{A}}_{uv}[d_u, d_v] : \forall d, d_u, d_v \in [k]\}\right), \tag{10}$$

where $[k] = \{1, \ldots, k\}$, $k$ is a receptive field, $\psi$ is an MLP and $\odot$ is the element-wise product. $p(u, v)$ decouples the readout function from subgraph generation as suggested in Figure 2b. This approach moves computation from edgewise operations e.g. generating subgraphs, to nodewise operations (generating sketches) that encapsulate the most relevant subgraph information for LP. We have shown that (i) GNN propagation is either not needed (structure features) or can be preprocessed globally (given features) (ii) structure features can be generated from sketches instead of subgraphs (iii) an edge pooling readout function performs as well as a graph pooling readout. In combination these approaches circumvent the need to explicitly construct subgraphs. The resulting model efficiently combines node features and the graph structure without explicitly constructing subgraphs and is as efficient as GCN (Kipf & Welling, 2017).

**Expressive power of ELPH**   ELPH combines advantageous traits of both GNNs and subgraph methods, i.e. tractable computational complexity and access to pair-wise structural features including triangle counts (Appendix C.5). Furthermore, ELPH is more expressive than MPNNs:

**Proposition 4.1.** *Let $\mathcal{M}_{\text{ELPH}}$ be the family of ELPH models as per Equations 7 - 9 and 10 where estimates are exact ($\hat{\mathcal{A}} \equiv \mathcal{A}, \hat{\mathcal{B}} \equiv \mathcal{B}$)[4]. $\mathcal{M}_{\text{ELPH}}$ does not suffer from the automorphic node problem.*

While we rigorously define the automorphic node problem and prove the above in Section A.2, this result states that there exists non-automorphic links with automorphic nodes that an ELPH model is able to discriminate (thanks to structural features). Contrary to ELPHs, MPNNs are not able to distinguish any of these links; we build upon this consideration, as well as the observation that ELPH models subsume MPNNs, to obtain this additional result:

**Theorem 4.2.** *Let $\mathcal{M}_{\text{MPNN}}$ be the family of Message Passing Neural Networks (Equation 1). $\mathcal{M}_{\text{ELPH}}$ is strictly more powerful than $\mathcal{M}_{\text{MPNN}}$ ($\mathcal{M}_{\text{ELPH}} \sqsubset \mathcal{M}_{\text{MPNN}}$).*

We prove this in Section A.2. Intuitively, the Theorem states that while all links separated by MPNNs are also separated by ELPHs, there also exist links separated by the latter family but not by the former.

## 5   SCALING ELPH WITH PREPROCESSING (BUDDY)

Similarly to other full-graph GNNs (e.g. GCN), ELPH is efficient when the dataset fits into GPU memory. When it does not, the graph must be batched into subgraphs. Batching is a major challenge associated with scalable GNNs and invariably introduces high levels of redundancy across batches. Here we introduce a large scale version of ELPH, called BUDDY, which uses preprocessing to side-step the need to have the full dataset in GPU memory.

**Preprocessing**   Figure 4b indicates that fixed and learnable SGNN propagation give almost equivalent performance. Fixed propagation can be achieved by efficient sparse scatter operations and done only once in preprocessing. Sketches can also be precomputed in a similar way:

$$\mathbf{M}^{(l)} = \text{scatter\_min}(\mathbf{M}^{(l-1)}, G), \ \mathbf{H}^{(l)} = \text{scatter\_max}(\mathbf{H}^{(l-1)}, G), \ \mathbf{X}^{(l)} = \text{scatter\_mean}(\mathbf{X}^{(l-1)}, G)$$

where $\text{scatter\_min}(\mathbf{M}^{(l)}, G)_u = \min_{v \in \mathcal{N}(u)} \mathbf{m}_v^{(l-1)}$ and scatter_max and scatter_mean are defined similarly, $\mathbf{X}^{(0)}$ are the original node features, and $\mathbf{M}^{(0)}$ and $\mathbf{H}^{(0)}$ are the *MinHashing* and *HyperLogLog* single node sketches respectively. Similarly to (Rossi et al., 2020), we concatenate features diffused at different hops to obtain the input node features: $\mathbf{Z} = \left[\mathbf{X}^{(0)} \| \mathbf{X}^{(1)} \| \ldots \| \mathbf{X}^{(k)}\right]$.

---

[4]For sufficiently large samples, this result can be extended to approximate counts via unbiased estimators.

**Link Predictor**    We now have $p(u, v) = \psi\left(\mathbf{z}_u^{(k)} \odot \mathbf{z}_v^{(k)}, \{\hat{\mathcal{B}}_{uv}[d], \hat{\mathcal{A}}_{uv}[d_u, d_v] : \forall\, d, d_u, d_v \in [k]\}\right)$,
where $\psi$ is an MLP and $\hat{\mathcal{A}}_{uv}$ and $\hat{\mathcal{B}}_{uv}$ are computed using $\mathbf{M}$ and $\mathbf{H}$ as explained in Section 4.1. Doing so effectively converts a GNN into an MLP, removing the need to sample batches of subgraphs when the dataset overflows GPU memory (Rossi et al., 2020). Further improvements in efficiency are possible when multiple computations on the same edge sets are required. In this case, the edge features $\{\hat{\mathcal{B}}_{uv}[d], \hat{\mathcal{A}}_{uv}[d_u, d_v]\}$ can be precomputed and cached.

**Time Complexity**    Denoting the complexity of hash operations as $h$, the node representation dimension $d$ and the number of edges $E = |\mathcal{E}|$, the preprocessing complexity of BUDDY is $\mathcal{O}(kEd + kEh)$. The first term being node feature propagation and the second sketch propagation. Preprocessing for SEAL and NBFNet are $\mathcal{O}(1)$. A link probability is computed by (i) extracting $k(k+2)$ $k$-hop structural features, which costs $\mathcal{O}(k^2 h)$ and (ii) An MLP on structural and node features, which results in $\mathcal{O}(k^2 h + kd^2)$ operations. Both are **independent of the size of the graph**. Since SGNNs construct subgraphs for each link they are $\mathcal{O}(Ed^2)$ for complex networks (See Section C.2). Finally, the total complexity of NBFNet is $\mathcal{O}(Ed + Nd^2)$ with an amortized time for a single link prediction of $\mathcal{O}(Ed/N + d^2)$ (Zhu et al., 2021). However, this is only realized in situations where the link probability $p(u, j)$ is required $\forall j \in \mathcal{V}$. Time complexities are summarized in Table 1 and space complexities in Appendix B.4.

**Expressiveness of BUDDY**    Similarly to ELPH, BUDDY does not suffer from the automorphic node problem (see Proposition and its Proof in Appendix A.2). However, due to its non-parametric feature propagation, this class of models does not subsume MPNNs and we cannot exclude the presence of links separated by MPNNs, but not by BUDDY. Nevertheless, BUDDY empirically outperforms common MPNNs by large margins (see Section 7), while also being extremely scalable.

# 6    RELATED WORK

Subgraph methods for link prediction were introduced as the Weisfeiler Leman Neural Machine (WLNM) in (Zhang & Chen, 2017). As an MLP is used to learn from subgraphs instead of a GNN their model is not permutation-equivariant and uses a hashing-based WL algorithm (Kerst-

Table 1: Model time complexity. $N$ and $E$ are the number of nodes and edges respectively. We use $d$-dimensional node features, $k$ hops for propagation and sketches of size $h$.

| Complexity | SEAL | NBFNet | BUDDY |
|---|---|---|---|
| Preprocessing | $\mathcal{O}(1)$ | $\mathcal{O}(1)$ | $\mathcal{O}(kE(d+h))$ |
| Training (1 link) | $\mathcal{O}(Ed^2)$ | $\mathcal{O}(Ed + Nd^2)$ | $\mathcal{O}(k^2 h + kd^2)$ |
| Inference | $\mathcal{O}(Ed^2)$ | $\mathcal{O}(Ed + Nd^2)$ | $\mathcal{O}(k^2 h + kd^2)$ |

ing et al., 2014) to generate node indices for the MLP. Note that 'hashing' here refers to injective neighbor aggregation functions, distinct from the data sketching methods employed in the current work. The MLP in WLNM was replaced by a GNN in the seminal SEAL (Zhang & Chen, 2018), thus removing the need for a hashing scheme. Additional methodological improvements were made in (Zhang et al., 2021; Zhang & Li, 2021) and subgraph generation was improved in (Yin et al., 2022). Applications to recommender systems were addressed in (Zhang & Chen, 2019) and random walks are used to represent subgraph topology in (Yin et al., 2022; Pan et al., 2022). The DRNL labeling scheme was introduced in SEAL and further labeling schemes are developed in (You et al., 2021; Li et al., 2020) and generalized with equivariant positional encodings (Wang et al., 2022). Neural Bellman-Ford Networks (Zhu et al., 2021) takes a different approach. It is concerned with single source distances where each layer of the neural networks is equivalent to an iteration of the Bellman-Ford algorithm and can be regarded as using a partial labeling scheme. The same graph and augmentation can be shared among multiple destinations and so it is faster (amortized) than SGNNs. However, it suffers from poor space complexity and thus requires impractically high memory consumption. Neo-GNN (Yun et al., 2021) directly learns a heuristic function using an edgewise and a nodewise neural network. #GNN (Wu et al., 2021) directly hashes node features, instead of graph structure, and is limited to node features that can be interpreted as set memberships. CFLP Zhao et al. (2022) applies causal inference to LP. Another paradigm for LP uses self-supervised node embeddings combined with an (often) approximate distance metric. Amongst the best known models are TransE (Bordes et al., 2013), DistMult (Yang et al., 2015), or complEx (Trouillon et al., 2016), which have been shown to scale to large LP systems in e.g. (Lerer et al., 2019; El-Kishky et al., 2022). Decoupling feature propagation and learning in GNNs was first implemented in SGC (Wu et al., 2019) and then scalably in SIGN (Rossi et al., 2020).

Table 2: Results on link prediction benchmarks. The top three models are colored by **First**, **Second**, **Third**. Where possible, baseline results are taken directly from the OGB leaderboard.

| Metric | Cora HR@100 | Citeseer HR@100 | Pubmed HR@100 | Collab HR@50 | PPA HR@100 | Citation2 MRR | DDI HR@20 |
|---|---|---|---|---|---|---|---|
| **CN** | $33.92_{\pm0.46}$ | $29.79_{\pm0.90}$ | $23.13_{\pm0.15}$ | $56.44_{\pm0.00}$ | $27.65_{\pm0.00}$ | $51.47_{\pm0.00}$ | $17.73_{\pm0.00}$ |
| **AA** | $39.85_{\pm1.34}$ | $35.19_{\pm1.33}$ | $27.38_{\pm0.11}$ | $64.35_{\pm0.00}$ | $32.45_{\pm0.00}$ | $51.89_{\pm0.00}$ | $18.61_{\pm0.00}$ |
| **RA** | $41.07_{\pm0.48}$ | $33.56_{\pm0.17}$ | $27.03_{\pm0.35}$ | $64.00_{\pm0.00}$ | $49.33_{\pm0.00}$ | $51.98_{\pm0.00}$ | $27.60_{\pm0.00}$ |
| **transE** | $67.40_{\pm1.60}$ | $60.19_{\pm1.15}$ | $36.67_{\pm0.99}$ | $29.40_{\pm1.15}$ | $22.69_{\pm0.49}$ | $76.44_{\pm0.18}$ | $6.65_{\pm0.20}$ |
| **complEx** | $37.16_{\pm2.76}$ | $42.72_{\pm1.68}$ | $37.80_{\pm1.39}$ | $53.91_{\pm0.50}$ | $27.42_{\pm0.49}$ | $72.83_{\pm0.38}$ | $8.68_{\pm0.36}$ |
| **DistMult** | $41.38_{\pm2.49}$ | $47.65_{\pm1.68}$ | $40.32_{\pm0.89}$ | $51.00_{\pm0.54}$ | $28.61_{\pm1.47}$ | $66.95_{\pm0.40}$ | $11.01_{\pm0.49}$ |
| **GCN** | $66.79_{\pm1.65}$ | $67.08_{\pm2.94}$ | $53.02_{\pm1.39}$ | $47.14_{\pm1.45}$ | $18.67_{\pm1.32}$ | $84.74_{\pm0.21}$ | $37.07_{\pm5.07}$ |
| **SAGE** | $55.02_{\pm4.03}$ | $57.01_{\pm3.74}$ | $39.66_{\pm0.72}$ | $54.63_{\pm1.12}$ | $16.55_{\pm2.40}$ | $82.60_{\pm0.36}$ | $53.90_{\pm4.74}$ |
| **Neo-GNN** | $80.42_{\pm1.31}$ | $84.67_{\pm2.16}$ | $73.93_{\pm1.19}$ | $62.13_{\pm0.58}$ | $49.13_{\pm0.60}$ | $87.26_{\pm0.84}$ | $63.57_{\pm3.52}$ |
| **SEAL** | $81.71_{\pm1.30}$ | $83.89_{\pm2.15}$ | $75.54_{\pm1.32}$ | $64.74_{\pm0.43}$ | $48.80_{\pm3.16}$ | $87.67_{\pm0.32}$ | $30.56_{\pm3.86}$ |
| **NBFnet** | $71.65_{\pm2.27}$ | $74.07_{\pm1.75}$ | $58.73_{\pm1.99}$ | OOM | OOM | OOM | $4.00_{\pm0.58}$ |
| **ELPH** | $87.72_{\pm2.13}$ | $93.44_{\pm0.53}$ | $72.99_{\pm1.43}$ | $66.32_{\pm0.40}$ | OOM | OOM | $83.19_{\pm2.12}$ |
| **BUDDY** | $88.00_{\pm0.44}$ | $92.93_{\pm0.27}$ | $74.10_{\pm0.78}$ | $65.94_{\pm0.58}$ | $49.85_{\pm0.20}$ | $87.56_{\pm0.11}$ | $78.51_{\pm1.36}$ |

# 7 EXPERIMENTS

**Datasets, Baselines and Experimental Setup** We report results for the most widely used Planetoid citation networks Cora (McCallum et al., 2000), Citeseer (Sen et al., 2008) and Pubmed (Namata et al., 2012) and the OGB link prediction datasets (Hu et al., 2020). We report results for: Three heuristics that have been successfully used for LP, Adamic-Adar (AA) (Adamic & Adar, 2003), Resource Allocation (RA) (Zhou et al., 2009) and Common Neighbors (CN); two of the most popular GNN architectures: Graph Convolutional Network (GCN) (Kipf & Welling, 2017) and GraphSAGE (Hamilton et al., 2017); the state-of-the-art link prediction GNNs Neo-GNN (Yun et al., 2021), SEAL (Zhang & Chen, 2018) and NBFNet (Zhu et al., 2021). Additional details on datasets and the experimental setup are included in Appendix B.1 and the code is public[5].

**Results** Results are presented in Table 2 with metrics given in the first row. Either ELPH or BUDDY achieve the best performance in five of the seven datasets, with SEAL being the closest competitor. Being a full-graph method, ELPH runs out of memory on the two largest datasets, while its scalable counterpart, BUDDY, performs extremely well on both. Despite ELPH being more general than BUDDY, there is no clear winner between the two in terms of performance, with ELPH outperforming BUDDY in three of the five datasets where we have results for both. Ablation studies for number of hops, sketching parameters and the importance of node and structure features are in Appendix D.2.

**Runtimes** Wall times are shown in Table 3. We report numbers for both the static mode of SEAL (all subgraphs are generated and labeled as a preprocessing step), and the dynamic mode (subgraphs are generated on the fly). BUDDY is orders of magnitude faster both in training and inference. In particular, Buddy is 200–1000× faster than SEAL in dynamic mode for training and inference on the Citation dataset. We also show GCN as a baseline. Further runtimes and a breakdown of preprocessing costs are in Appendix D.1 with learning curves in Appendix F.

Table 3: Wall times. Training time is one epoch. Inference time is the full test set.

| dataset | time (s) | SEAL dyn | SEAL stat | ELPH | BUDDY | GCN |
|---|---|---|---|---|---|---|
| Pubmed | preproc | 0 | 630 | 0 | 5 | 0 |
| | train | 70 | 30 | 25 | 1 | 4 |
| | inference | 23 | 9 | 0.1 | 0.06 | 0.02 |
| Citation | preproc | 0 | $\sim 3,000,000$ | | 1,200 | |
| | train | $\sim300,000$ | $\sim200,000$ | OOM | 1,500 | OOM |
| | inference | $\sim300,000$ | $\sim100,000$ | | 300 | |

# 8 CONCLUSION

We have presented a new model for LP that is based on an analysis of existing state-of-the-art models, but which achieves better time and space complexity and superior predictive performance on a range of standard benchmarks. The current work is limited to undirected graphs or directed graphs that are first preprocessed to make them undirected as is common in GNN research. We leave as future work extensions to directed graphs and temporal / dynamically evolving graphs and investigations into the links with graph curvature (See Appendix C.5).

---

[5]https://github.com/melifluos/subgraph-sketching

## 9  ACKNOWLEDGMENTS

We would like to thank Junhyun Lee, Seongjun Yun and Michael Galkin for useful discussions regarding prior works and we are particularly grateful to Junhyun and Seongjun for generating additional Neo-GNN results for Table 2. MB is supported in part by ERC Consolidator grant no 724228 (LEMAN).

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

## A  THEORETICAL ANALYSES

### A.1  PRELIMINARIES

We introduce some preliminary concepts that will be useful in our analysis. Let us start with the definition of graph *isomorphism* and *automorphism*.

**Definition A.1** (Graph isomorphism and automorphism)**.** *Let $G_1 = (V_1, E_1)$, $G_2 = (V_2, E_2)$ be two simple graphs. An* isomorphism *between $G_1, G_2$ is a bijective map $\varphi : V_1 \to V_2$ which preserves adjacencies, that is: $\forall u, v \in V_1 : (u, v) \in E_1 \Longleftrightarrow (\varphi(u), \varphi(v)) \in E_2$. If $G_1 = G_2$, $\varphi$ is called an* automorphism*.*

In view of the definition above, two graphs are also called *isomorphic* whenever there exists an isomorphism between the two. Intuitively, if two graphs are isomorphic they encode the same exact relational structure, up to relabeling of their nodes. As we shall see next, automorphisms convey a slightly different meaning: they allow to define graph symmetries by formalizing the concept of node structural roles.

The above definitions can be extended to attributed graphs, more conveniently represented through third-order tensors $\mathsf{A} \in \mathbb{R}^{n^2 \times d}$, with $n = |V|, d > 0$ the number of features (Maron et al., 2019; Zhang et al., 2021). Indexes in the first two dimensions of tensor $\mathsf{A}$ univocally correspond to vertexes in $V$ through a bijection $\iota : V \to \{1, \ldots, n\}$. Here, elements $\big(\mathsf{A}\big)_{i,i,:}$ correspond to node features, while elements $\big(\mathsf{A}\big)_{i,j,:}, i \neq j$ to edge features, which include the connectivity information in $E$. Within this context, a graph is defined as a tuple $G = (V, E, \iota, \mathsf{A})$, and isomorphisms and automorphisms are interpreted as permutations acting on $\mathsf{A}$ as:

$$\big(\sigma \cdot \mathsf{A}\big)_{ijk} = \mathsf{A}_{\sigma^{-1}(i)\sigma^{-1}(j)k}, \quad \forall \sigma \in S_n \tag{11}$$

where $S_n$ is the set of all bijections (permutations) over $n$ symbols. We then define the concepts of isomorphism and automorphism as:

**Definition A.2** (Graph isomorphism and automorphism on attributed graphs)**.** *Let $G_1, G_2$ be two attributed graphs represented by tensors $\mathsf{A}_1, \mathsf{A}_2 \in \mathbb{R}^{n^2 \times d}$. An* isomorphism *between $G_1, G_2$ is a bijective map (permutation) $\sigma \in S_n$ such that $\sigma \cdot \mathsf{A}_1 = \mathsf{A}_2$. For a graph $G = \mathsf{A}$, $\sigma \in S_n$ is an* automorphism *if $\sigma \cdot \mathsf{A} = \mathsf{A}$.*

Let us get back to the information conveyed by automorphisms by introducing the following definition:

**Definition A.3** (Graph automorphism group)**.** *Let $G$ be a simple graph and $\mathbb{A}_G$ be the set of graph automorphisms defined on its vertex set. $\mathrm{Aut}_G = (\mathbb{A}_G, \circ)$ is the group having $\mathbb{A}_G$ as base set and function composition $(\circ)$ as its operation.*

Group $\mathrm{Aut}_G$ induces an equivalence relation $\sim_G \subseteq V \times V$: $\forall u, v \in V, u \sim_G v \Leftrightarrow \exists \varphi \in \mathbb{A}_G : u = \varphi(v)$ (it is easily proved that $\sim_G$ is indeed reflexive, symmetric and transitive, thus qualifying as an equivalence relation). The equivalence classes of $\sim_G$ partition the vertex set into *orbits*: $\mathrm{Orb}_G(v) = \{u \in V \mid u \sim_G v\} = \{u \in V \mid \exists \varphi \in \mathbb{A}_G : u = \varphi(v)\}$. The set of orbits identifies all structural roles in the graph. Two vertexes belonging to the same orbit have the same structural role in the graph and are called 'symmetric' or 'automorphic'.

In fact, it is possible to define automorphisms between node-pairs as well (Srinivasan & Ribeiro, 2019; Zhang et al., 2021):

**Definition A.4** (Automorphic node pairs)**.** *Let $G = \mathsf{A}$ be an attributed graph and $\ell_1 = (u_1, v_1), \ell_2 = (u_2, v_2) \in V \times V$ be two pairs of nodes. We say $\ell_1, \ell_2$ are* automorphic *if there exists an automorphism $\sigma \in \mathbb{A}_G$ such that $\sigma \cdot \ell_1 = (\sigma(u_1), \sigma(v_1)) = (u_2, v_2) = \ell_2$. We write $\ell_1 \sim_G^{(2)} \ell_2$.*

With the above it is possible to extend the definition of orbits to node-pairs, by considering those node-pair automorphisms which are naturally induced by node ones, i.e.: $\forall \varphi \in \mathbb{A}_G, \varphi^* : (u, v) \mapsto (\varphi(u), \varphi(v))$.

Notably, for two node pairs $\ell_1 = (u_1, v_1), \ell_2 = (u_2, v_2) \in V \times V$, $\{u_1 \sim_G u_2, v_1 \sim_G v_2\} \not\Longrightarrow \ell_1 \sim_G^{(2)} \ell_2$, i.e., automorphism between nodes does not imply automorphism between node pairs. For instance, one counter-example of automorphic nodes involved in non-automorphic pairs is depicted in Figure 1, while another one is constituted by pair $(v_0, v_2), (v_0, v_3)$ in Figure 7, whose automorphisms

are reported in Tables 4 and 5. This 'phenomenon' is the cause of the so-called "automorphic node problem".

**Definition A.5** (Automorphic node problem). *Let $\mathcal{M}$ be a family of models. We say $\mathcal{M}$ suffers from the automorphic node problem if for any model $M \in \mathcal{M}$, and any simple (attributed) graph $G = (V, E, \iota, \mathsf{A})$ we have that $\forall (u_1, v_1), (u_2, v_2) \in V \times V, \{u_1 \sim_G u_2, v_1 \sim_G v_2\} \implies M\big((u_1, v_1)\big) = M\big((u_2, v_2)\big)$.*

The above property is identified as a 'problem' because there exist examples of non-automorphic node pairs composed by automorphic nodes. A model with this property would inevitably compute the same representations for these non-automorphic pairs, despite they feature significantly different characteristics, e.g. shortest-path distance or number of common neighbors.

Importantly, as proved by Srinivasan & Ribeiro (2019) and restated by Zhang et al. (2021), the model family of Message Passing Neural Networks suffer from the aforementioned problem:

**Proposition A.6.** *Let $\mathcal{M}_{\mathrm{MPNN}}$ be the family of Message Passing Neural Networks (Equation 1) representing node-pairs as a function of their computed (equivariant) node representations. $\mathcal{M}_{\mathrm{MPNN}}$ suffers from the automorphic node problem.*

We conclude these preliminaries by introducing the concept of *link discrimination*, which gives a (more) fine-grained measure of the link representational power of model families.

**Definition A.7** (Link discrimination). *Let $G = (V, E, \iota, \mathsf{A})$ be any simple (attributed) graph and $M$ a model belonging to some family $\mathcal{M}$. Let $\ell_1 = (u_1, v_1), \ell_2 = (u_2, v_2) \in V \times V$ be two node pairs. We say $M$ discriminates pairs $\ell_1, \ell_2$ iff $M(\ell_1) \neq M(\ell_2)$. We write $\ell_1 \neq_M \ell_2$. If there exists such a model $M \in \mathcal{M}$, then family $\mathcal{M}$ distinguishes between the two pairs and we write $\ell_1 \neq_{\mathcal{M}} \ell_2$.*

Accordingly, $\mathcal{M}$ does not discriminate pairs $\ell_1, \ell_2$ when it contains no model instances which assign distinct representations to the pairs. We write $\ell_1 =_{\mathcal{M}} \ell_2$.

We can compare model families based on their *expressiveness*, that is, their ability to discriminate node pairs:

**Definition A.8** (More expressive). *Let $\mathcal{M}_1, \mathcal{M}_2$ be two model families. We say $\mathcal{M}_1$ is more expressive than $\mathcal{M}_2$ iff $\forall G = (V, E, \iota, \mathsf{A}), \ell_1, \not\sim_G^{(2)} \ell_2 \in V \times V, \ell_1 \neq_{\mathcal{M}_2} \ell_2 \implies \ell_1 \neq_{\mathcal{M}_1} \ell_2$. We write $\mathcal{M}_1 \sqsubseteq \mathcal{M}_2$.*

Put differently, $\mathcal{M}_1$ is more expressive than $\mathcal{M}_2$ when for any two node pairs, if there exists a model in $\mathcal{M}_2$ which disambiguates between the two pairs, then there exists a model in $\mathcal{M}_1$ which does so as well. When the opposite is not verified, we say $\mathcal{M}_1$ is *strictly* more expressive than $\mathcal{M}_2$:

**Definition A.9** (Strictly more expressive). *Let $\mathcal{M}_1, \mathcal{M}_2$ be two model families. We say $\mathcal{M}_1$ is strictly more expressive than $\mathcal{M}_2$ iff $\mathcal{M}_1 \sqsubseteq \mathcal{M}_2 \wedge \mathcal{M}_2 \not\sqsubseteq \mathcal{M}_1$. Equivalently, $\mathcal{M}_1 \sqsubseteq \mathcal{M}_2 \wedge \exists G = (V, E, \iota, \mathsf{A}), \ell_1, \not\sim_G^{(2)} \ell_2 \in V \times V$, s.t. $\ell_1 \neq_{\mathcal{M}_1} \ell_2 \wedge \ell_1 =_{\mathcal{M}_2} \ell_2$.*

In other words, $\mathcal{M}_1$ is strictly more expressive than $\mathcal{M}_2$ when there exists no model in the latter family disambiguating between two non-automorphic pairs while there exist some models in the former which do so.

## A.2 DEFERRED THEORETICAL RESULTS AND PROOFS

Let us start by reporting the Proof for Proposition 4.1, stating that ELPH models do not suffer from the automorphic node problem.

*Proof of Proposition 4.1.* In order to prove the Proposition it is sufficient to exhibit an ELPH model which distinguishes between two node-pairs whose nodes are automorphic. Consider $G = C_6$, the chordless cycle graph with 6 nodes, which we depict in Figure 7. Due to the symmetry of this graph, all nodes are in the same orbit, and are therefore automorphic: we report the set of all graph automorphisms $\mathbb{A}_G$ in Table 4. Let us then consider pairs $\ell_1 = (v_0, v_2), \ell_2 = (v_0, v_3)$. The two pairs satisfy the premise in the definition of the automorphic node problem, as $v_0 \sim_G v_0, v_2 \sim_G v_3$. One single ELPH message-passing layer ($k = 1$) produces the following (exact) structural features. Pair $\ell_1$: $\mathcal{A}_{v_0,v_2}[1,1] = 1, \mathcal{B}_{v_0,v_2}[1] = 1$; pair $\ell_2$: $\mathcal{A}_{v_0,v_3}[1,1] = 0, \mathcal{B}_{v_0,v_3}[1] = 2$. Thus, a one-layer ELPH model $M$ is such that $\ell_1 \neq_M \ell_2$ if the readout layer $p(u, v) = \psi\left(\mathbf{x}_u^1 \odot \mathbf{x}_v^1, (\hat{\mathcal{B}}_{uv}[1], \hat{\mathcal{A}}_{uv}[1,1])\right) =$

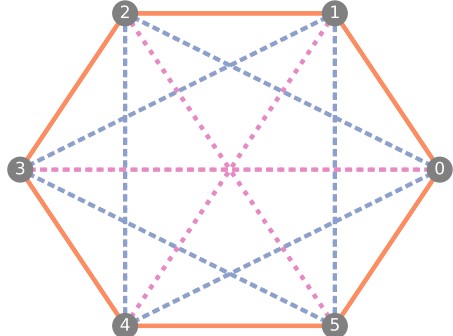

Figure 7: Graph $C_6$ with nodes and node-pairs coloured according to the orbit they belong to. Node-pairs corresponding to actual edges are depicted solid, dashed otherwise. There are $2 \cdot n = 12$ automorphisms which map any node to any other (see Table 4). Hence all nodes are in the same, single orbit. On the contrary, node-pairs are partitioned into three distinct orbits (see Table 5). As it is possible to notice, pairs $(v_0, v_2)$ and $(v_0, v_3)$ are *not* automorphic, while their constituent nodes are.

Table 4: Images, for each node, according to the 12 automorphisms for graph $C_6$. See Figure 7.

| vertex | $A_1$ | $A_2$ | $A_3$ | $A_4$ | $A_5$ | $A_6$ | $A_7$ | $A_8$ | $A_9$ | $A_{10}$ | $A_{11}$ | $A_{12}$ |
|---|---|---|---|---|---|---|---|---|---|---|---|---|
| 0 | 0 | 0 | 1 | 1 | 2 | 2 | 3 | 3 | 4 | 4 | 5 | 5 |
| 1 | 1 | 5 | 0 | 2 | 1 | 3 | 2 | 4 | 3 | 5 | 0 | 4 |
| 2 | 2 | 4 | 5 | 3 | 0 | 4 | 1 | 5 | 2 | 0 | 1 | 3 |
| 3 | 3 | 3 | 4 | 4 | 5 | 5 | 0 | 0 | 1 | 1 | 2 | 2 |
| 4 | 4 | 2 | 3 | 5 | 4 | 0 | 5 | 1 | 0 | 2 | 3 | 1 |
| 5 | 5 | 1 | 2 | 0 | 3 | 1 | 4 | 2 | 5 | 3 | 4 | 0 |

$\hat{\mathcal{A}}_{uv}[1,1]$. An MLP implementing the described $\psi$ is only required to nullify parts of its input and apply an identity mapping on the remaining ones. This MLP trivially exists (it can be even manually constructed). □

We now move to the counterpart of the above result for BUDDY models.

**Proposition A.10.** *Let* $\mathcal{M}_{\text{BUDDY}}$ *be the family of BUDDY models as described per Equations 10 and pre-processed features* $\mathbf{Z} = \left[ \mathbf{X}^{(0)} \parallel \mathbf{X}^{(1)} \parallel ... \parallel \mathbf{X}^{(k)} \right]$, *where estimates are exact* $(\hat{\mathcal{A}} \equiv \mathcal{A}, \hat{\mathcal{B}} \equiv \mathcal{B})$. $\mathcal{M}_{\text{BUDDY}}$ *does not* suffer from the automorphic node problem.

*Proof of Proposition A.10.* In order to prove the Proposition it is sufficient to notice that the readout in the above Proof completely neglects node features, while only replicating in output the structural features $\mathcal{A}$. This is also a valid BUDDY readout function, and allows BUDDY to output the same exact node-pair representations of the ELPH model described above. As we have observed, these are enough to disambiguate $(v_0, v_2), (v_0, v_3)$ from graph $C_6$ depicted in Figure 7. This concludes the proof. □

In these results we have leveraged the assumption that $\mathcal{A}$ estimates are exact. We observe that, if the MinHash and HyperLogLog estimators are unbiased, it would be possible to choose a sample size large enough to provide distinct estimates for the two counts of interest, so that the above results continue to hold. Unbiased cardinality estimators are, for instance, described in (Ertl, 2017).

A more precise assessment of the representational power of ELPH can be obtained by considering the node-pairs this model family is able to discriminate w.r.t. others.

**Lemma A.11.** *Let* $\mathcal{M}_{\text{ELPH}}$ *be the family of ELPH models (Equations 7, 8, 9, and 10),* $\mathcal{M}_{\text{MPNN}}$ *that of Message Passing Neural Networks (Equation 1).* $\mathcal{M}_{\text{ELPH}}$ *is more powerful than* $\mathcal{M}_{\text{MPNN}}$ $(\mathcal{M}_{\text{ELPH}} \sqsubseteq \mathcal{M}_{\text{MPNN}})$.

Table 5: Images, for each node-pair, according to the 12 induced node-pair automorphisms for graph $C_6$. Pairs are considered as undirected (sets). See Figure 7. As it is possible to notice, there is no automorphism mapping pair $(v_0, v_2)$ to $(v_0, v_3)$ – hence the two are in distinct orbits.

| pair | $A_1$ | $A_2$ | $A_3$ | $A_4$ | $A_5$ | $A_6$ | $A_7$ | $A_8$ | $A_9$ | $A_{10}$ | $A_{11}$ | $A_{12}$ |
|---|---|---|---|---|---|---|---|---|---|---|---|---|
| (0, 1) | (0, 1) | (0, 5) | (0, 1) | (1, 2) | (1, 2) | (2, 3) | (2, 3) | (3, 4) | (3, 4) | (4, 5) | (0, 5) | (4, 5) |
| (0, 2) | (0, 2) | (0, 4) | (1, 5) | (1, 3) | (0, 2) | (2, 4) | (1, 3) | (3, 5) | (2, 4) | (0, 4) | (1, 5) | (3, 5) |
| (0, 3) | (0, 3) | (0, 3) | (1, 4) | (1, 4) | (2, 5) | (2, 5) | (0, 3) | (0, 3) | (1, 4) | (1, 4) | (2, 5) | (2, 5) |
| (0, 4) | (0, 4) | (0, 2) | (1, 3) | (1, 5) | (2, 4) | (0, 2) | (3, 5) | (1, 3) | (0, 4) | (2, 4) | (3, 5) | (1, 5) |
| (0, 5) | (0, 5) | (0, 1) | (1, 2) | (0, 1) | (2, 3) | (1, 2) | (3, 4) | (2, 3) | (4, 5) | (3, 4) | (4, 5) | (0, 5) |
| (1, 2) | (1, 2) | (4, 5) | (0, 5) | (2, 3) | (0, 1) | (3, 4) | (1, 2) | (4, 5) | (2, 3) | (0, 5) | (0, 1) | (3, 4) |
| (1, 3) | (1, 3) | (3, 5) | (0, 4) | (2, 4) | (1, 5) | (3, 5) | (0, 2) | (0, 4) | (1, 3) | (1, 5) | (0, 2) | (2, 4) |
| (1, 4) | (1, 4) | (2, 5) | (0, 3) | (2, 5) | (1, 4) | (0, 3) | (2, 5) | (1, 4) | (0, 3) | (2, 5) | (0, 3) | (1, 4) |
| (1, 5) | (1, 5) | (1, 5) | (0, 2) | (0, 2) | (1, 3) | (1, 3) | (2, 4) | (2, 4) | (3, 5) | (3, 5) | (0, 4) | (0, 4) |
| (2, 3) | (2, 3) | (3, 4) | (4, 5) | (3, 4) | (0, 5) | (4, 5) | (0, 1) | (0, 5) | (1, 2) | (0, 1) | (1, 2) | (2, 3) |
| (2, 4) | (2, 4) | (2, 4) | (3, 5) | (3, 5) | (0, 4) | (0, 4) | (1, 5) | (1, 5) | (0, 2) | (0, 2) | (1, 3) | (1, 3) |
| (2, 5) | (2, 5) | (1, 4) | (2, 5) | (0, 3) | (0, 3) | (1, 4) | (1, 4) | (2, 5) | (2, 5) | (0, 3) | (1, 4) | (0, 3) |
| (3, 4) | (3, 4) | (2, 3) | (3, 4) | (4, 5) | (4, 5) | (0, 5) | (0, 5) | (0, 1) | (0, 1) | (1, 2) | (2, 3) | (1, 2) |
| (3, 5) | (3, 5) | (1, 3) | (2, 4) | (0, 4) | (3, 5) | (1, 5) | (0, 4) | (0, 2) | (1, 5) | (1, 3) | (2, 4) | (0, 2) |
| (4, 5) | (4, 5) | (1, 2) | (2, 3) | (0, 5) | (3, 4) | (0, 1) | (4, 5) | (1, 2) | (0, 5) | (2, 3) | (3, 4) | (0, 1) |

*Proof of Lemma A.11.* We prove this Lemma by noticing that the ELPH architecture generalizes that of an MPNN, so that an ELPH model can learn to simulate a standard MPNN by ignoring the structural features. Specifically, ELPH defaults to an MPNN with (1) $\phi^{(l)}\left(\mathbf{x}_u^{(l-1)}, \mathbf{x}_v^{(l-1)}, \mathbf{e}_{u,v}^{(l)}\right) = \phi^{(l)}\left(\mathbf{x}_u^{(l-1)}, \mathbf{x}_v^{(l-1)}\right)$, and (2) $\psi\left(\mathbf{x}_u^k \odot \mathbf{x}_v^k, \{\hat{\mathcal{B}}_{uv}[d], \hat{\mathcal{A}}_{uv}[d_u, d_v] : \forall\, d, d_u, d_v = 1, \ldots, k\}\right) = \psi\left(\mathbf{x}_u^k \odot \mathbf{x}_v^k\right)$. This entails that, anytime a specific MPNN instance distinguishes between two non-automorphic node-pairs, there exists an ELPH model which does so as well: the one which exactly simulates such MPNN. □

**Lemma A.12.** *There exist node-pairs distinguished by an ELPH or BUDDY model (with exact cardinality estimates) which are not distinguished by any MPNN model.*

*Proof of Lemma A.12.* The Lemma is proved simply by considering non-automorphic pairs $(v_0, v_2), (v_0, v_3)$ from graph $C_6$ depicted in Figure 7. We have already shown how there exists both an ELPH and BUDDY model (computing exact estimates of $\mathcal{A}, \mathcal{B}$) which separate the two. On the other hand, we have already observed how the nodes in the pairs are automorphic as all belonging to the same orbit. Thus, the two pairs cannot possibly be distinguished by any MPNN, since they suffer from the automorphic node problem (Srinivasan & Ribeiro, 2019; Zhang et al., 2021), as remarked in Proposition A.6. □

*Proof of Theorem 4.2.* The Theorem follows directly from Lemmas A.11 and A.12. □

Does this expressiveness result also extend to BUDDY? While we have proved BUDDY does not suffer from the automorphic node problem, its non-parametric message-passing scheme is such that this family of models do not generally subsume MPNNs. On one hand, in view of Lemma A.12, we know there exist node-pairs separated by BUDDY but not by any MPNN; on the other, this does not exclude the presence of node-pairs for which the vice-versa is true.

## B ADDITIONAL EXPERIMENTAL DETAILS

### B.1 DATASETS AND THEIR PROPERTIES

Table 6 basic properties of the experimental datasets, together with the scaling of subgraph size with hops. Subgraph statistics are generated by expanding $k$-hop subgraphs around 1000 randomly selected links. Regular graphs scale as $\deg^k$, however as these datasets are all complex networks the size of subgraphs grows far more rapidly than this, which poses serious problems for SGNNs. Furthermore, the size of subgraphs is highly irregular with high standard deviations making efficient

Table 6: Properties of link prediction benchmarks. Confidence intervals are $\pm$ one standard deviation. Splits for the Planetoid datasets are random and Collab uses the fixed OGB splits. Where possible, baseline results for Collab are taken directly from the OGB leaderboard

|  | **Cora** | **Citeseer** | **Pubmed** | **Collab** | **PPA** | **DDI** | **Citation2** |
|---|---|---|---|---|---|---|---|
| #Nodes | 2,708 | 3,327 | 18,717 | 235,868 | 576,289 | 4,267 | 2,927,963 |
| #Edges | 5,278 | 4,676 | 44,327 | 1,285,465 | 30,326,273 | 1,334,889 | 30,561,187 |
| splits | rand | rand | rand | time | throughput | time | protein |
| avg $deg$ | 3.9 | 2.74 | 4.5 | 5.45 | 52.62 | 312.84 | 10.44 |
| avg $deg^2$ | 15.21 | 7.51 | 20.25 | 29.70 | 2769 | 97,344 | 109 |
| 1-hop size | $12\pm15$ | $8\pm8$ | $12\pm17$ | $99\pm251$ | $152\pm152$ | $901\pm494$ | $23\pm28$ |
| 2-hop size | $127\pm131$ | $58\pm92$ | $260\pm432$ | $115\pm571$ | $7790\pm6176$ | $3830\pm412$ | $285\pm432$ |

parallelization in scalable architectures challenging. The one exception to this pattern is DDI. Due to the very high density of DDI, most two-hop subgraphs include almost every node.

### B.2 EXPERIMENTAL SETUP

In all cases, we use the largest connected component of the graph. LP tasks require links to play dual roles as both supervision labels and message passing links. For all datasets, at training time the message passing links are equal to the supervision links, while at test and validation time, disjoint sets of links are held out for supervision that are never seen at training time. The test supervision links are also never seen at validation time, but for the Planetoid and ogbl-collab[6] datasets, the message passing edges at test time are the union of the training message passing edges and the validation supervision edges. OGB datasets have fixed splits whereas for Planetoid, random 70-10-20 percent train-val-test splits were generated. On DDI, NBFnet was trained without learned node embeddings and with a batch size of 5.

### B.3 HYPERPARAMETERS

The $p$ parameter used by *HyperLogLog* was 8 and the number of permutations used by *MinHashing* was 128. All hyperparameters were tuned using Weights and Biases random search. The search space was over hidden dimension (64–512), learning rate (0.0001–0.01) and dropout (0–1), layers (1–3) and weight decay (0–0.001). Hyperparameters with the highest validation accuracy were chosen and results are reported on a test set that is used only once.

### B.4 SPACE COMPLEXITY

The space complexity for ELPH is almost the same as a GCN based link predictor. We define the number of feature as $F$, the sketch size $H$ (sum of hll and minhash sketch sizes), number of nodes as $N$, the number of layers $L$, the batch size $B$ and the number of edges as $E$. We split link prediction into i) learning node representations and ii) predicting link probabilities from node representations. Learning node representations with GCN has complexity

$$E + LF^2 + LNF, \tag{12}$$

where the second and third terms represent the weight matrices and the node representation respectively. ELPH has complexity,

$$E + LF^2 + LN(F + H) \tag{13}$$

And so the only difference is the additional space required to store the node sketches. The GCN link predictor has space complexity

$$BF + F^2, \tag{14}$$

Where the first term is a batch of edges and the second term are the link predictor weight matrices. ELPH also uses structure features, which assuming the number of GNN layers and the number of hops are the same, take up L(L+2) space per node. The link prediction complexity of ELPH is then

$$B(F + L(L + 2)) + F^2 + F(L(L + 2)). \tag{15}$$

---

[6]The OGB rules allow validation edges to be used for the ogbl-collab dataset.

Typically the number of layers L is 1,2,3 and so the additional overhead is small. The space complexity of the BUDDY link predictor is the same as ELPH and node representations are built as a preprocessing step. The preprocessing has space complexity

$$E + LNF + LN + LNH, \tag{16}$$

which correspond to the adjacency matrix, the propagated node features, the hll cardinality estimates and the minhash and hll sketches respectively. The only difference between this and ELPH is that cardinality estimates are cached (LN) and no weight matrices are used.

### B.5 Implementation Details

Our code is implemented in PyTorch (Paszke et al., 2019), using PyTorch geometric (Fey & Lenssen, 2019). Code and instructions to reproduce the experiments are available at `https://github.com/melifluos/subgraph-sketching`. We utilized either AWS p2 or p3 machines with 8 Tesla K80 and 8 Tesla V100 respectively to perform all the experiments in the paper.

## C More on Structural Features

### C.1 Heuristic Methods for Link Prediction

Heuristic methods are classified by the receptive field and whether they measure neighborhood similarity or path length. The simplest neighborhood similarity method is the 1-hop Common Neighbor (CN) count. Many other heuristics such as cosine similarity, the Jaccard index and the Probabilistic Mutual Information (PMI) differ from CN only by the choice of normalization. Adamic-Adar and Resource Allocation are two closely related second order heuristics that penalize neighbors by a function of the degree

$$\Gamma(u, v) = \sum_{i \in N(u) \cap N(v)} \frac{1}{f(|N(i)|)}, \tag{17}$$

where $N(u)$ are the neighbors of $u$. Shortest path based heuristics are generally more expensive to calculate than neighborhood similarity as they require global knowledge of the graph to compute exactly. The Katz index takes into account multiple paths between two nodes. Each path is given a weight of $\alpha^d$, where $\alpha$ is a hyperparameter attenuation factor and $d$ is the path length. Similarly Personalized PageRank (PPR) estimates landing probabilities of a random walker from a single source node. For a survey comparing 20 different LP heuristics see e.g. (Lü & Zhou, 2011).

### C.2 Subgraph Generation Complexity

The complexity for generating regular $k$-hop subgraph is $\mathcal{O}(deg^k)$, where $deg$ is the degree of every node in the subgraph. For graphs that are approximately regular, with a well defined mean degree, the situation is slightly worse. However, in general we are interested in complex networks (such as social networks, recommendation systems or citation graphs) that are formed as a result of preferential attachment. Complex networks are typified by power law degree distributions of the form $p(deg) = deg^{-\gamma}$. As a result the mean is only well defined if $\gamma > 2$ and the variance is finite only for $\gamma > 3$. As most real world complex networks fall into the range $2 < \gamma < 3$, we have that the maximum degree is only bounded by the number of edges in the graph and thus, so is the complexity of subgraph generation. Table 6 includes the average size of one thousand 1-hop and 2-hop randomly generated graphs for each dataset used in our experiments. In all cases, the size of subgraphs greatly exceeds the average degree baseline with very large variances.

### C.3 Subgraph Sketches

We make use of both the *HyperLogLog* and *MinHash* sketching schemes. The former is used to estimate the size of the union, while the latter estimates the Jaccard similarity between two sets. Together they can be used to estimate the size of set intersections.

#### C.3.1 HyperLogLog

HyperLogLog efficiently estimates the cardinality of large sets. It accomplishes this by representing sets using a constant size data sketch. These sketches can be combined in time that is constant w.r.t

the data size and linear in the sketch size using elementwise maximum to estimate the size of a set union.

The algorithm takes the precision $p$ as a parameter. From $p$, it determines the number of registers to use. The sketch for a set $S$ is comprised of $m$ registers $M_1 \ldots M_m$ where $m = 2^p$. A hash function $h(s)$ maps elements from $S$ into an array of 64-bits. The algorithm uses the first $p$ bits of the hash to associate elements with a register. From the remaining bits, it computes the number of leading zeros, tracking the maximum number of leading zeros per register. Intuitively a large number of leading zero bits is less likely and indicates a higher cardinality and for a single register the expected cardinality for a set where the maximum number of leading zeros is $n$ is $2^n$. This estimate is highly noisy and there are several methods to combine estimates from different registers. We use hyperloglog++ (Heule et al., 2013), for which the standard error is numerically close to $1.04/sqrt(m)$ for large enough $m$.

Hyperloglog can be expressed in three functions Initialize, Union, and Card. Sketches are easily merged by populating a new set of registers with the element-wise max values for each register. To extract the estimate, the algorithm finds the harmonic mean of $2^{M[m]}$ for each of the the $m$ registers. This mean estimates the cardinality of the set divided by $m$. To find the estimated cardinality we multiply by $m$ and $\alpha_m$. $\alpha_m$ is used to correct multiplicative bias. Additional information about the computation of $\alpha_m$ along with techniques to improve the estimate can be found in (Flajolet et al., 2007; Heule et al., 2013). The full algorithm is presented in Algorithm 1.

**Complexity** The Initialize operation has a $\mathcal{O}(m + |S|)$ running time. Union and Card are both $\mathcal{O}(m)$ operations. The size of the sketch for a 64-bit hash is $6 * 2^p$ bits.

### C.3.2 MINHASHING

The MinHash algorithm estimates the Jaccard index. It can similarly be expressed in three functions Initialize, Union, and $J$. The $p$ parameter is the number of permutations on each element of the set. $P_i(x)$ computes a permutation of input $x$ where each $i$ specifies a different permutation. The algorithm stores the minimum value for each of the $p$ permutations of all hashed elements. The Jaccard estimate of the similarity of two sets is given by the Hamming similarity of their sketches. The full algorithm is presented as Algorithm 2.

**Complexity** The Initialize operation has a $\mathcal{O}(np|S|)$ running time. Union and $J$ are both $\mathcal{O}(np)$ operations. The size of the sketch is $np$ longs. Minhashing gives an unbiased estimate of the Jaccard with a variance given by the Cramer-Rao lower bound that scales as $\mathcal{O}(1/np)$ (Chamberlain et al., 2018).

### C.4 LABELING SCHEMES

Empirically, we found the best performing labeling scheme to be DRNL, which scores each node in $S_{uv}$ based on it's distance to $u$ and $v$ with the caveat that when scoring the distance to $u$, node $v$ and all of its edges are masked and vice versa. This improves expressiveness as otherwise for positive edges every neighbor of $u$ would always be a 2-hop neighbor of $v$ and vice-versa. The result of masking is that distances can be very large even for 2-hop graphs (for instance one node may have an egonet that is a star graph with a ring around the outside. If the edges of the star are masked then the ring must be traversed). The first few DRNL values are

1. disconnected nodes: $(\infty, 0), (0, \infty) \rightarrow 0$
2. link nodes: $(0, 1), (1, 0) \rightarrow 1$
3. common neighbors: $(1, 1) \rightarrow 2$
4. 12-common-neighbors $(1, 2), (2, 1) \rightarrow 3$
5. 2-hop common neighbors: $(2, 2) \rightarrow 4$

and the pattern has a hash function given by

$$f_l(i) = 1 + \min(d_{ui}, d_{vi} + (d/2)[(d//2) + d\%2) - 1] \tag{18}$$

where $d = d_{ui} + d_{vi}$. It is slightly suboptimal to assign infinite distances to 0, which is at least part of the reason that they are one-hot encode them as labels. Indeed, DE (Li et al., 2020) uses a max distance label, which they claim reduces overfitting.

---

**Algorithm 1** HyperLogLog: Estimate cardinality

---

Parameter $p$ is used to control precision.
$m = 2^p$
**procedure** HLLINITIALIZE($S$)
    **for** $i \in range(m)$ **do**
        $M[i] = 0$
    **end for**
    **for** $v \in S$ **do**
        $x = h(v)$
        $idx = \langle x_{31}, \ldots, x_{32-p}\rangle_2$
        $w = \langle x_{31-p}, \ldots, x_0\rangle_2$
        $M[idx] = max(M[idx], \varrho(w)))$
    **end for**
    **return** $M$
**end procedure**

**procedure** HLLUNION($M1$, $M2$)
    **for** $i \in range(m)$ **do**
        $M[i] = max(M1[i], M2[i]))$
    **end for**
    **return** $M$
**end procedure**

**procedure** CARD($M$)
    **return** $\alpha_m m^2 (\sum_{i=0}^{m} 2^{-M[i]})^{-1}$
**end procedure**

---

### C.5 SUBSTRUCTURE COUNTING AND CURVATURE

The eight features depicted in Figure 6 can be used to learn local substructures. $\mathcal{A}_{uv}[1,1]$ counts the number of triangles that $(u,v)$ participates in assuming that the edge $(u,v)$ exists. $\mathcal{A}_{uv}[2,1]$ and $\mathcal{A}_{uv}[1,2]$ will double count a four-cycle and single count a four-cycle with a single diagonal. The model can not distinguish a four-clique from two triangles or a five-clique from three triangles. $\mathcal{A}_{uv}[2,2]$ counts five cycles. LP has also recently been shown to be closely related to notions of discrete Ricci curvature on graphs (Topping et al., 2022), a connection that we plan exploring in future work.

### C.6 EXAMPLE STRUCTURE FEATURES

Figure 8 provides an example of how structure features are calculated from a subgraph. The figure provides the $z$ values for each node in an eight node subgraph. From these the structure features are calculated. As an example

$$\mathcal{A}_{67}[2,1] = |\{2,3,4,6,7\} \cap \{2,8\}| = |\{2\}| = 1 \tag{19}$$
$$\mathcal{B}_{67}[2] = |\{2,3,4,6,7\} \setminus \{1,2,3,5,6,7,8\}| = |\{4\}| = 1 \tag{20}$$

The value of $\mathcal{A}_{67}[2,1] = 1$ indicates that the is one node in common between the two-hop neighbors of node 6 and the 1-hop neighbors of node 7. The common node is node 2. Similarly $\mathcal{B}_{67}[2] = 1$ means there is one element in the two-hop neighbors of 6 that is not in any of the k-hop neighbors of 7: node 4.

## D ADDITIONAL EXPERIMENTS

### D.1 RUNTIMES AND DISCUSSION

The results in Tables 3 and 7 are obtained by running methods on a single Tesla K80 GPU on an AWS p2 machine. In all cases SEAL used GCN (fastest GNN) and parameters were taken from the

---

**Algorithm 2** MinHash: Estimate Jaccard Similarity

---

Parameter $np$ controls the number of permutations.
**procedure** MINHASHINITIALIZE($S$)
    **for** $i \in range(np)$ **do**
        $M[i] = maxvalue$
    **end for**
    **for** $v \in S$ **do**
        $x = h(v)$
        **for** $i \in range(np)$ **do**
            $M[i] = min(M[i], P_i(x))$
        **end for**
    **end for**
    **return** $M$
**end procedure**

**procedure** MINHASHUNION($M1$, $M2$)
    **for** $i \in range(np)$ **do**
        $M[i] = min(M1[i], M2[i]))$
    **end for**
    **return** $M$
**end procedure**

**procedure** J($M1$, $M2$)
    $num\_equal = 0$
    **for** $i \in range(np)$ **do**
        **if** $M1[i] = M2[i]$ **then**
            $num\_equal + +$
        **end if**
    **end for**
    **return** $num\_equal/np$
**end procedure**

---

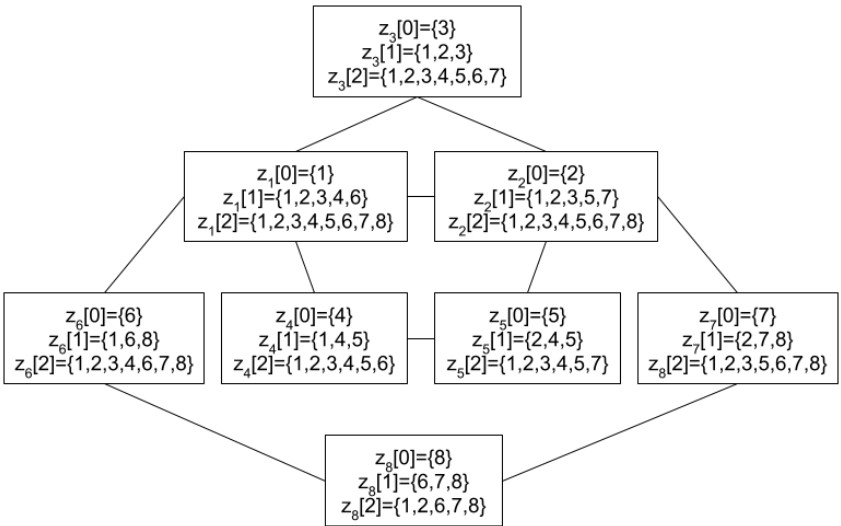

Figure 8: Computed z features

Table 7: Wall time of our methods in comparison to SEAL in dynamic and static mode. Training time is for one epoch. Inference time is for the full test set. Due to high runtimes, values for SEAL are estimated from samples for the OGB datasets

| dataset | Wall time (sec) | SEAL dyn | SEAL stat | ELPH | Buddy |
|---------|-----------------|----------|-----------|------|-------|
| Cora | Preprocessing | 0 | 57 | 0 | 0.4 |
| | Training (1 epoch) | 9 | 5 | 2 | 0.7 |
| | Inference | 3 | 2 | 0.05 | 0.04 |
| Citeseer | Preprocessing | 0 | 48 | 0 | 0.7 |
| | Training (1 epoch) | 13 | 5 | 1 | 0.5 |
| | Inference | 20 | 2 | 0.03 | 0.07 |
| Pubmed | Preprocessing | 0 | 630 | 0 | 5 |
| | Training (1 epoch) | 70 | 32 | 25 | 1 |
| | inference | 23 | 9 | 0.1 | 0.06 |
| Collab | Preprocessing | 0 | ~25000 | 0 | 43 |
| | Training (1 epoch) | ~5,000 | ~330 | 2100 | 105 |
| | Inference | ~5,000 | ~160 | 2 | 1 |
| PPA | Preprocessing | 0 | ~900,000 | | 840 |
| | Training (1 epoch) | ~130,000 | ~150,000 | OOM | 75 |
| | Inference | ~20,000 | ~16,000 | | 18 |
| Citation | Preprocessing | 0 | ~ 3,000,000 | | 1,200 |
| | Training (1 epoch) | ~300,000 | ~200,000 | OOM | 1,450 |
| | inference | ~300,000 | ~100,000 | | 280 |
| DDI | Preprocessing | 0 | ~400,000 | 0 | 66 |
| | Training (1 epoch) | ~150,000 | ~250,000 | 30 | 27 |
| | Inference | ~1,000 | ~19,000 | 0.6 | 0.4 |

Table 8: The three types of preprocessing used in BUDDY and associated wall times. Entity gives the entity associated with the preprocessed features. Each node has a hash and propagated features while each edge requires structure features

| Wall time (sec) | Entity | Cora | Citeseer | Pubmed | Collab | PPA | Citation | DDI |
|-----------------|--------|------|----------|--------|--------|-----|----------|-----|
| hashing | node | 0.12 | 0.1 | 1.04 | 26 | 469 | 714 | 20.3 |
| feature propagation | node | 0.27 | 0.60 | 0.67 | 4.6 | 77 | 126 | NA |
| structure features | edge | 0.02 | 0.01 | 3.3 | 12.0 | 294 | 393 | 45.76 |

SEAL OGB repo. For the OGB datasets, SEAL runtimes are estimated based on samples due to the high runtimes.

Table 8 breaks BUDDY preprocessing times down into (i) generating hashes and (ii) propagating features for each node, where the same values are used for training and inference and (iii) constructing structure features from hashes for each query edge, which has a separate cost for inference. We stress that both forms of preprocessing depend only on the dataset and so must only be done once ever for a fixed dataset and not e.g. after every epoch. This is akin to the static behavior of SEAL (Zhang & Chen, 2018), where subgraphs are constructed for each edge (both training and inference) as a preprocessing step.

## D.2    ABLATION STUDIES

This section contains ablation studies for (i) the affect of varying the number of *Minhash* permutations and the *HyperLogLog* $p$ parameter and (ii) removing either node features or structure features from BUDDY.

### D.2.1    HASHING PARAMETER ABLATION

Figures 9 and 10 are ablation studies of the hashing parameters that trade-off between accuracy and time/space complexity of the intersection estimates. The method is relatively insensitive to both parameters allowing smaller values to be chosen when space / time complexity is a constraint. Figure 9a shows that good performance is achieved providing more than 16 minhash permutations

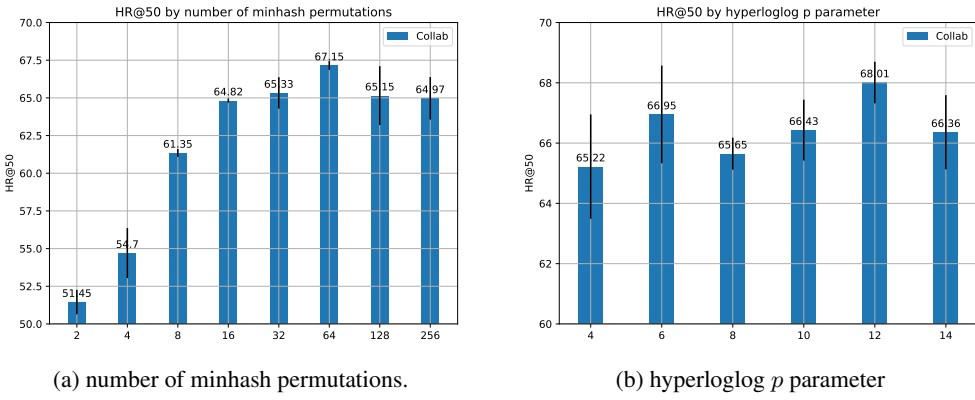

(a) number of minhash permutations.

(b) hyperloglog $p$ parameter

Figure 9: Ablation study for hashing parameters for Collab dataset

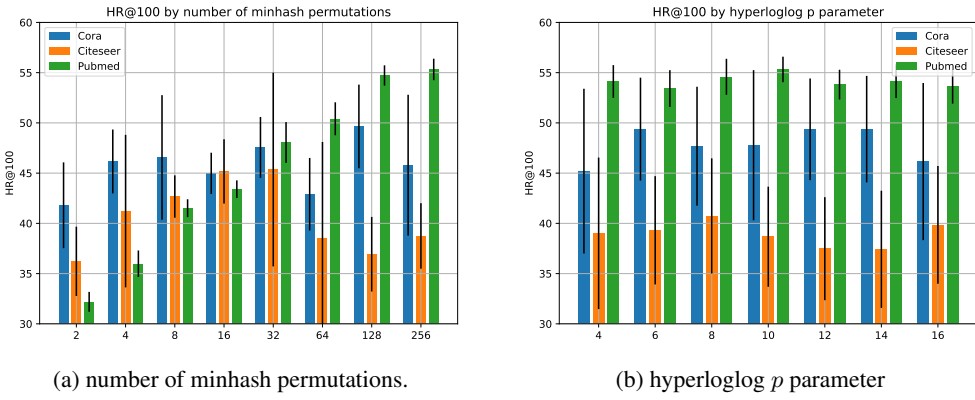

(a) number of minhash permutations.

(b) hyperloglog $p$ parameter

Figure 10: Ablation study for hashing parameters for Planetoid datasets

are used, while Figure 9b shows that $p$ can be as low as 4 in the hyperloglog procedure. For Figure 10 values were calculated with no node features to emphasize the affect of only the hashing parameters. This was not required for Figure 9 because relatively speaking the node features are less important for Collab (See Table 9).

Data sketching typically introduces a tradeoff between estimation accuracy and time and space complexity. However, in our model, the time complexity for generating structure features from hashes is negligible at both training and inference time compared to the cost of a forward pass of the MLP.

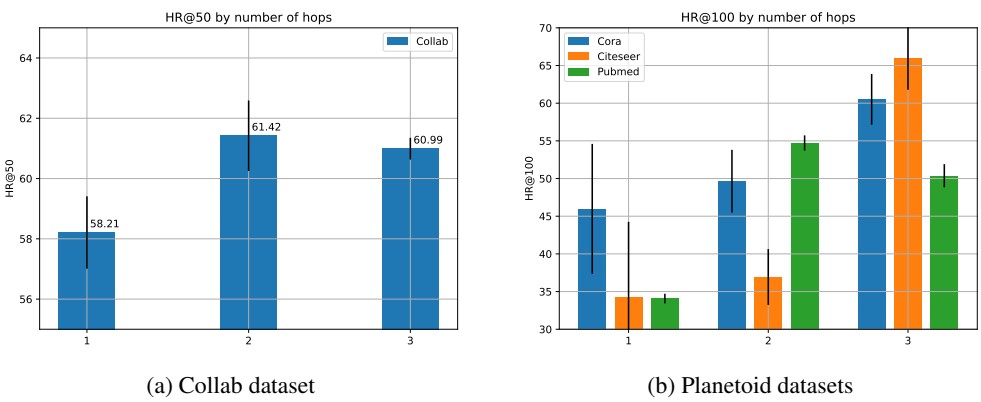

(a) Collab dataset

(b) Planetoid datasets

Figure 11: Ablation study for the number of hops

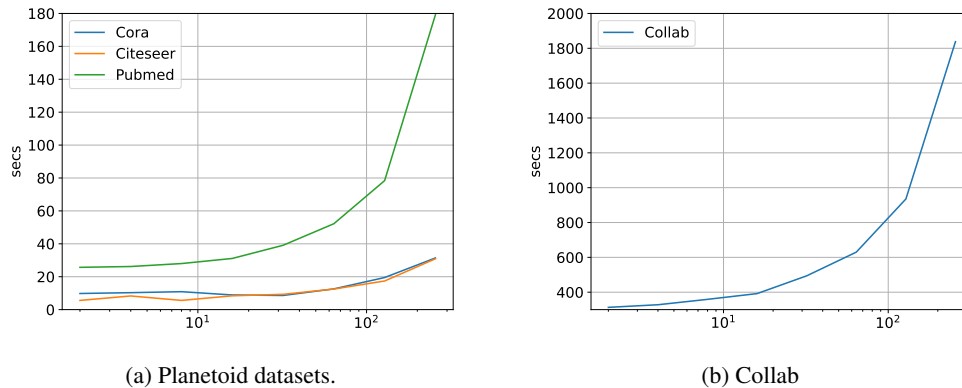

(a) Planetoid datasets.

(b) Collab

Figure 12: runtime of hash generation against number of minhash permutations

|  | Cora | Citeseer | Pubmed | Collab | PPA | Citation2 | DDI |
|---|---|---|---|---|---|---|---|
| #Nodes | 2,708 | 3,327 | 18,717 | 235,868 | 576,289 | 2,927,963 | 4267 |
| #Edges | 5,278 | 4,676 | 44,327 | 1,285,465 | 30,326,273 | 30,561,187 | 1,334,889 |
| avg deg | 3.9 | 2.74 | 4.5 | 5.45 | 52.62 | 10.44 | 312.84 |
| metric | HR@100 | HR@100 | HR@100 | HR@50 | HR@100 | MRR | HR@20 |
| CN | $33.92_{\pm0.46}$ | $29.79_{\pm0.90}$ | $23.13_{\pm0.15}$ | $56.44_{\pm0.00}$ | $27.65_{\pm0.00}$ | $51.47_{\pm0.00}$ | $17.73_{\pm0.00}$ |
| AA | $39.85_{\pm1.34}$ | $35.19_{\pm1.33}$ | $27.38_{\pm0.11}$ | $64.35_{\pm0.00}$ | $32.45_{\pm0.00}$ | $51.89_{\pm0.00}$ | $18.61_{\pm0.00}$ |
| RA | $41.07_{\pm0.48}$ | $33.56_{\pm0.17}$ | $27.03_{\pm0.35}$ | $64.00_{\pm0.00}$ | $49.33_{\pm0.00}$ | $51.98_{\pm0.00}$ | $27.60_{\pm0.00}$ |
| **BUDDY** | **$88.00_{\pm0.44}$** | **$92.93_{\pm0.27}$** | **$74.10_{\pm0.78}$** | **$65.94_{\pm0.58}$** | **$49.85_{\pm0.20}$** | **$87.56_{\pm0.11}$** | **$78.51_{\pm1.36}$** |
| w\0 Features | $48.45_{\pm4.83}$ | $36.33_{\pm5.59}$ | $53.50_{\pm2.23}$ | $60.46_{\pm0.33}$ | $49.85_{\pm0.20}$ | $82.27_{\pm0.10}$ | NA |
| w\0 SF | $83.90_{\pm2.28}$ | $91.24_{\pm1.44}$ | $65.57_{\pm2.86}$ | $22.83_{\pm1.26}$ | $1.20_{\pm0.21}$ | $83.59_{\pm0.13}$ | $74.01_{\pm13.18}$ |

Table 9: Ablation table showing the affects of removing both structure features and node features from BUDDY with all hyperparameters held fixed. Core heuristics are shown for comparison with the w\o features row. Confidence intervals are $\pm$ one sd. Planetoid splits are random and the OGB splits are fixed.

The only place where these parameters have an appreciable impact on runtimes is in preprocessing the hashes. When preprocessing the hashes, the cost of generating hyperloglog sketches is also negligible compared to the cost of the minhash sketches. The relationship between the number of minhash permutations and the runtime to generate the hashes is shown in Figure 12.

### D.2.2 FEATURE ABLATION

Table 9 shows the degradation in performance of BUDDY with either structure features or node features removed with all hyperparameters held fixed. DDI has no node features and BUDDY did not use the node features from PPA, which are one-hot species labels. For datasets Collab and PPA, the structure features dominate performance. For the Citation dataset, the contribution of structure features and node features is almost equal, with a relatively small incremental benefit of adding a second feature class. For the Planetoid datasets adding structure features gives a significant, but relatively small incremental benefit beyond the node features. It should also be noted that combining node and structure features dramatically reduces the variance over runs for the Planetoid and Collab datasets.

## E FULL BUDDY ALGORITHM

A sketch of the full algorithm is given in Algorithm 3

## F LEARNING CURVES

We provide learning curves in terms of number of epochs in Figures 13 - 19.

---

**Algorithm 3** Complete Procedure

Preprocess structure features and cache propagated node features with Graph $G$ and features $X$
**procedure** PREPROCESSING(G, X)
    $H1 = \text{MINHASHINITIALIZE}(G)$
    $H2 = \text{HLLINITIALIZE}(G)$
    $X' = Propagate(X)$
**end procedure**
Generate edge probability predictions $y$ using an MLP
**procedure** PREDICT(H1,H2,X')
    **for** edge $(u, v) \in$ epoch **do**
        $SF_{u,v} = GetStructureFeatures(u, v, H1, H2)$
        $x_u = GetNodeFeatures(u, X')$
        $x_v = GetNodeFeatures(v, X')$
        $y = \text{MLP}(SF_{u,v}, x_u, x_v)$
    **end for**
**end procedure**

---

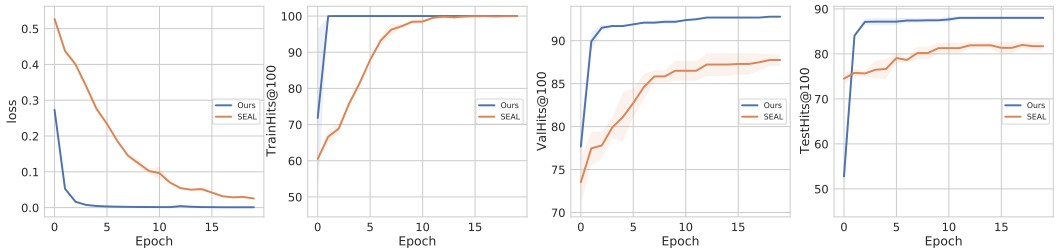

Figure 13: Loss and train-val-test learning curves as a function of training epoch, averaged over restarts. Solid line represents mean value and shadowed region shows one standard deviation. Cora dataset.

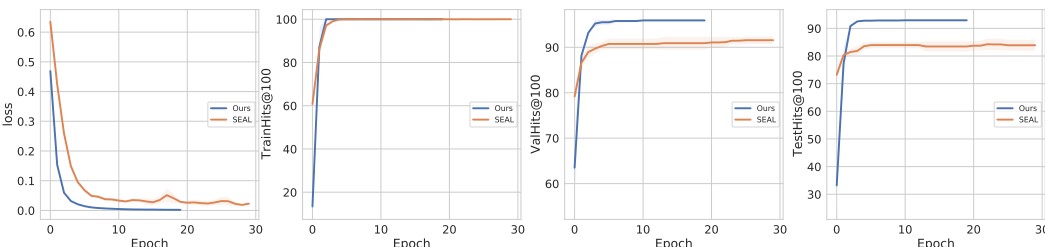

Figure 14: Loss and train-val-test learning curves as a function of training epoch, averaged over restarts. Solid line represents mean value and shadowed region shows one standard deviation. Citeseer dataset.

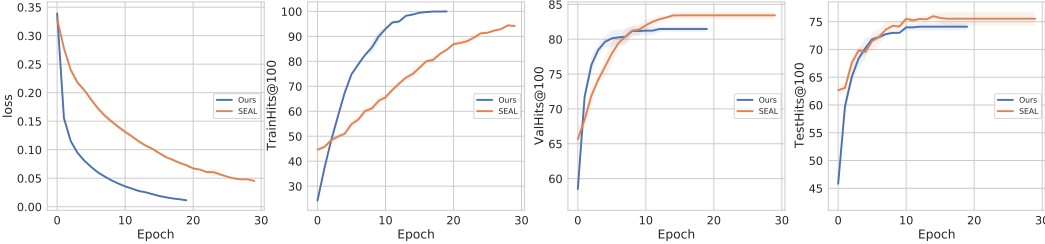

Figure 15: Loss and train-val-test learning curves as a function of training epoch, averaged over restarts. Solid line represents mean value and shadowed region shows one standard deviation. Pubmed dataset.

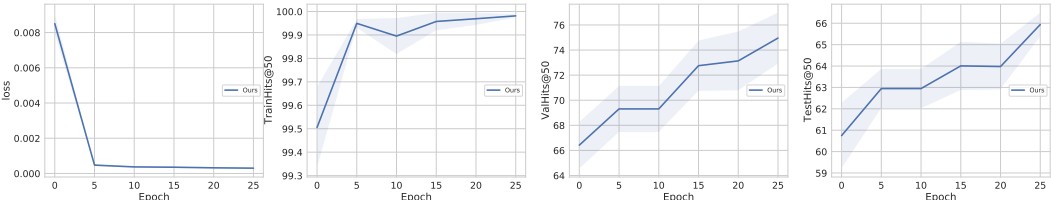

Figure 16: Loss and train-val-test learning curves as a function of training epoch, averaged over restarts. Solid line represents mean value and shadowed region shows one standard deviation. ogbl-Collab dataset.

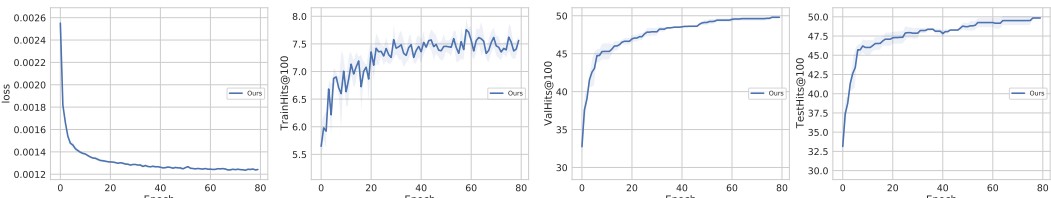

Figure 17: Loss and train-val-test learning curves as a function of training epoch, averaged over restarts. Solid line represents mean value and shadowed region shows one standard deviation. ogbl-PPA dataset.

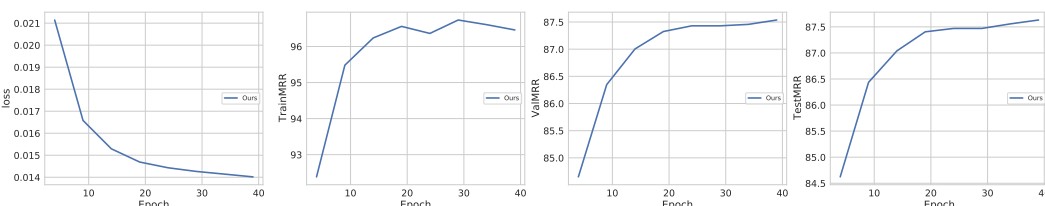

Figure 18: Loss and train-val-test learning curves as a function of training epoch, averaged over restarts. Solid line represents mean value and shadowed region shows one standard deviation. ogbl-Citation2 dataset.

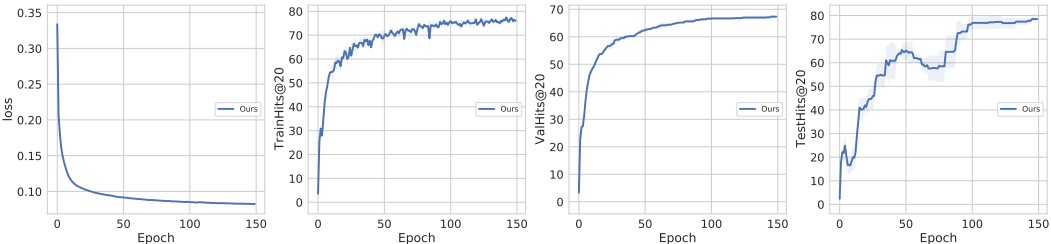

Figure 19: Loss and train-val-test learning curves as a function of training epoch, averaged over restarts. Solid line represents mean value and shadowed region shows one standard deviation. ogbl-DDI dataset.

# G  SOCIETAL IMPACT

We study LP in graph-structured datasets focusing primarily on methods rather than applications. Our method, in principle, may be employed in industrial recommendation systems. We have no evidence that our method enhances biases, but were it to be deployed, checks would need to be put in place that existing biases were not amplified.