# OpenReview forum: "Graph Neural Networks for Link Prediction with Subgraph Sketching"
_ICLR.cc/2023/Conference — ICLR 2023 notable top 5%_

### Official Review · Reviewer_ucmP · 2022-10-23

**Confidence:** 3
**Correctness:** 4
**Technical Novelty And Significance:** 3
**Empirical Novelty And Significance:** 3
**Recommendation:** 8

**Clarity, Quality, Novelty And Reproducibility:**

The paper is well-organized and well-written. The proposed method is motivated by careful experimental studies and clearly described. The idea that passes subgraph sketches as messages seems novel to me.

**Strength And Weaknesses:**

Strength:
1. The analysis on the components of existing SGNN methods for link prediction is well-conducted and the results are informative, which may inspire new thoughts on SGNN-based link prediction.
2. The idea that passes subgraph sketches as messages is novel.
3. The empirical results render the efficiency and effectiveness of the proposed method.
4. A more scalable variant is implemented, which prepropagates the node features in the preprocessing stage. This further mitigate the scalability issue of SGNN-based link prediction such as SEAL.

Weaknesses:
I found the expressiveness analysis is less exciting. Here, more powerful means the capability of distinguishing automorphic nodes. But this can be realized by simple techniques such as adding random features. So, I don't think this is a good theoretical justification of better performance. Can you provide some comments on this?

**Summary Of The Paper:**

This paper studies GNN-based link prediction models. The authors experimentally analyze the components of existing subgraph GNN (SGNN) methods for link prediction, which exhibit some limitations and redundancy. Then propose a novel method that passes subgraph sketches as messages, which mitigates these issues effectively.

**Summary Of The Review:**

The paper investigates the components of existing SGNN link prediction methods and identifies several key limitations and opportunities of improvements. Then proposes a novel method to use subgraph sketches as messages. Empirical results clearly show the advantages of the proposed method.

---

### Official Review · Reviewer_jqUA · 2022-10-24

**Confidence:** 4
**Correctness:** 4
**Technical Novelty And Significance:** 4
**Empirical Novelty And Significance:** 4
**Recommendation:** 8

**Clarity, Quality, Novelty And Reproducibility:**

The presentation of the paper is clear and easy to follow. Also, the proposed idea is very interesting and novel. See the above section on strengths and weaknesses.


**Strength And Weaknesses:**

Strengths

- The essential factors for the performance of existing subgraph-level GNNs are well analyzed through the experimental observation of the SGNNs.

- The proposed idea of effectively estimating intersections and cardinalities without subgraph construction is interesting and novel.
By leveraging structural features, the proposed method addresses the inefficiency of existing subgraph-based methods and the automorphic node problem in MPNN.

- Experiments conducted on six link prediction benchmarks show both the effectiveness and efficiency of ELPH and BUDDY compared to recent GNN-based baselines.

Weakness

- Performance between baselines in ogbl-collab does not seem to be compared fairly. The performance of SEAL in ogbl-collab seems to be the performance using the validation set for training whereas other GNN-based methods (GCN, SAGE, Neo-GNN) don’t seem to use the validation set for training.

- It would be good if there were comparisons with GNNs with existing proposed structural features (e.g., DE [1] and node2vec [2]).

[1] Li, Pan, et al. "Distance encoding: Design provably more powerful neural networks for graph representation learning." Advances in Neural Information Processing Systems (2020)

[2] Grover, Aditya, and Jure Leskovec. "node2vec: Scalable feature learning for networks." Proceedings of the 22nd ACM SIGKDD international conference on Knowledge discovery and data mining. 2016.

**Summary Of The Paper:**

This paper presents a novel Graph Neural Networks, ELPH and BUDDY, that estimates key structural information in subgraph GNN without explicit subgraph construction. Specifically, the proposed method estimates intersection and cardinalities using subgraph sketching and incorporates them into Message Passing type GNNs. The paper provably shows more expressive power than Message Passing GNNs and experiments conducted on six link prediction benchmarks show both effectiveness and efficiency compared to state-of-the-art baselines.


**Summary Of The Review:**

This paper presents a novel Graph Neural Networks, ELPH and BUDDY, that estimates key structural information in subgraph GNN without explicit subgraph construction. The presentation of the paper is clear and the proposed idea is very interesting and novel.

---

### Official Review · Reviewer_9q6R · 2022-10-25

**Confidence:** 4
**Correctness:** 3
**Technical Novelty And Significance:** 3
**Empirical Novelty And Significance:** Not applicable
**Recommendation:** 8

**Clarity, Quality, Novelty And Reproducibility:**

Clarity: the paper is overall clearly written and easy to follow.

Quality: the proposed method seems sound to me.

Novelty: the proposed method is novel by combining the advantages of two approaches.

Reproducibility: while the description of the proposed method is detailed, the implementation was not provided.

**Strength And Weaknesses:**

S1. This paper is overall clearly written and easy to follow.

S2. The proposed method is very well motivated with comprehensive analysis on why subgraph GNNs works well.

S3. The proposed method is able to successfully combine the advantages of SGNN as well as normal GNNs for link prediction.

W1. Part of the notations in Sec. 2 are a bit confusing. For example. is $Z_{uv}$ a matrix / set of node embeddings or a vector of subgraph embedding? And why is the $u$ in $\mathbf{y}_{\mathbf{u}}$ (Eq. (1)) bolded, but not bolded elsewhere (e.g., in the following paragraph)?
I suggest the authors to follow the common way of denoting vectors and matrices. E.g., using bold lowercase for vectors and bold uppercase for matrices. Currently most but no all of the paper follows that.

W2. I assume the complexity in Sec. 5 and Table 1 are time complexity. If yes, it would be nice if the authors can also provide the space complexity analysis.

W3. As one of the goal of the proposed method is to obtain the effectiveness of subgraph GNNs while still being efficient and scalable, I think it would make the evaluation more comprehensive if the authors can also include the runtime of normal GNNs in Table 3 for comparison.

Minor ones:

W4. Missing some recent relevant link prediction citations. E.g., [1][2]

W5. In the first paragraph of Introduction, why are points i and ii bolded by not iii?

[1] Learning from Counterfactual Links for Link Prediction, ICML 2022 \
[2] Neural Link Prediction with Walk Pooling, ICLR 2022

**Summary Of The Paper:**

This paper focused on the research problem of link prediction with GNNs. The authors comprehensively analyzed why the subgraph-based GNNs performs good on link prediction, as well as their efficiency problems. The authors then proposed to estimate the key structural information with subgraph sketching. Doing so combined the advantages of effectiveness on SGNNs as well as the efficiency on full-graph GNNs.


**Summary Of The Review:**

Based on the above comments, I recommend acceptance. But I still encourage the authors to address the weaknesses.

---

### Official Review · Reviewer_ntmc · 2022-10-31

**Confidence:** 4
**Clarity, Quality, Novelty And Reproducibility:** The overall writing quality is very g…
**Correctness:** 3
**Technical Novelty And Significance:** 4
**Empirical Novelty And Significance:** 4
**Recommendation:** 10

**Strength And Weaknesses:**

I am the author of one of the main baselines compared in the paper. I believe that this is an excellent paper with the following significant contributions:
1. It unifies the several existing state-of-the-art methods for link prediction prediction and does systematic analysis to show their actual strengths of design.
2. It is fascinating to see how the computational redundancy in subgraph-based neighbor counting can be eliminated by the message passing form of graph sketch using hyperloglog and minhashing. This is one of the most innovative and smartest GNN designs I have seen in a while.
3. The experiments including those in Appendix D show that the proposed method perform very well by both accuracy and speed. The main claims in the theoretical parts are well substantiated by the experiments.

Weaknesses:
I don't see it as a strong negative but I have a question about Fig. 2: it is a bit surprising to see that DRNL (SEAL) can outperforms DE by such a huge margin on Pubmed, considering that SEAL's node labeling function is not more expressive that DE's one-hot distance encodings. Can any explanation be provided on that?
Also just a minor typo: the citation after "Distance Encoding (DE)" on Page 4 seems incorrect.

**Summary Of The Paper:**

This paper proposes a faster and more powerful GNN dedicated for the link prediction task. It captures the essence of success from previous state-of-the-art link prediction methods: subgraph-based structure encoding, which is essentially counts of various "common neighbors". To accelerate subgraph-based structure encoding, the authors sharply draw from its connection with the message passing form of computing graph sketches, and integrate this process into MPNN's framework. Experiments show that the proposed method achieves better performance compared to previous SOTAs in terms of both accuracy and scalability.



**Summary Of The Review:**

I strongly recommend the acceptance of the paper for reasons outlined above. However, it would be great to see the code be made available considering that this is very likely to be an impactful piece of research.

---

### Decision · Program_Chairs · 2023-01-20

**Decision:**

Accept: notable-top-5%

**Justification For Why Not Higher Score:**

N/A

**Justification For Why Not Lower Score:**

A novel method to solve an important problem with good performance and theoretical justification.

**Metareview: Summary, Strengths And Weaknesses:**

Graph neural networks and heuristics are two popular types of methods for link prediction. However, due to limitations in expressive power, GNNs cannot distinguish certain important patterns which heuristics can. To solve this problem, the authors develop an GNN that passes subgraph sketches as messages, which allow important qualities of the subgraphs to be summarized in the nodes. Theoretical analysis shows the proposed method is strictly more expressive, as it solves the automorphic node problem. Extensive experiments, include OGB leaderboard results, are included to show the promising performance of the method.

Strength
- Good review and analysis of existing works.
- A novel method to solve an important problem with good performance and theoretical justification.

Weakness
- Running time analysis (fix after revision).

Others, encourage fixing in the final version
- Construction of h^(0)_u and m^(0)_u, what are dimension size, and what min/max operation mean (operate on a scalar or vector)?
- The abstract has talked about "counting triangles" and "automorphic nodes". In the main text, the authors have talked about "automorphic nodes", how about "counting triangles"?
- Could the authors add the synthetic experiments to compare different methods' capabilities on distinguish "automorphic nodes"?

**Note From Pc:**

if the above contains the word "oral" or "spotlight" please see: "oral" presentation means -> notable-top-5% and "spotlight" means -> notable-top-25%. As stated in our emails, we are disassociating presentation type from AC recommendations

---

> ### Author Response · Authors · 2023-02-28
> **Thank you and responses to meta-review**
>
> Dear AC and reviewing team,
>
> Thank you again for your positive opinion of our paper and your valuable comments that have allowed us to improve the manuscript. Below we answer the remaining questions from the meta-review:
>
> > Running time analysis (fix after revision).
>
> This has been added to the camera-ready paper
>
> > Construction of h^(0)_u and m^(0)_u, what are dimension size, and what min/max operation mean (operate on a scalar or vector)?
>
> min and max are elementwise. The operations are described in detail in Appendix C3 and the main text has been adapted to make it clear that these are elementwise operations. The sizes are hyperparameters. Hyperparameters are discussed in Appendix C3, with the specific values given in Appendix B3.
>
> > The abstract has talked about "counting triangles" and "automorphic nodes". In the main text, the authors have talked about "automorphic nodes", how about "counting triangles"?
>
> The triangle counts are somewhat trivial as for an edge (u,v) they are the intersection of the 1-hop neighbors of u and v, which is the value of A_{uv}[1,1] in Figure 6. The only complexity is that we use an unbiased estimate (from the hashes) of this value and not the exact count. This is discussed in the "structure features" paragraph in Section 3 and in Appendix  C5. We have added a reference to the appendix just above proposition 4.1.
>
> > Could the authors add the synthetic experiments to compare different methods' capabilities on distinguish "automorphic nodes"?
>
> Thank you for this excellent suggestion. We believe that this synthetic experiment and a number of additional experiments relating to node features would provide new perspectives on the automorphic node problem and that this will be the basis for follow on work.